# Sampling-based Bayesian inference in recurrent circuits of stochastic spiking neurons

Wen-Hao Zhang ®[1,2,3,4,11], Si Wu[5,6,7,8], Krešimir Josić ®[9,10,12] ✉ & Brent Doiron ®[1,2,3,4,12] ✉

Two facts about cortex are widely accepted: neuronal responses show large spiking variability with near Poisson statistics and cortical circuits feature abundant recurrent connections between neurons. How these spiking and circuit properties combine to support sensory representation and information processing is not well understood. We build a theoretical framework showing that these two ubiquitous features of cortex combine to produce optimal sampling-based Bayesian inference. Recurrent connections store an internal model of the external world, and Poissonian variability of spike responses drives flexible sampling from the posterior stimulus distributions obtained by combining feedforward and recurrent neuronal inputs. We illustrate how this framework for sampling-based inference can be used by cortex to represent latent multivariate stimuli organized either hierarchically or in parallel. A neural signature of such network sampling are internally generated differential correlations whose amplitude is determined by the prior stored in the circuit, which provides an experimentally testable prediction for our framework.

In an uncertain and changing world, it is imperative for the brain to reliably represent and interpret external stimuli. The cortex is essential for the representation of the sensory world, and it is believed that populations of neurons collectively code for richly structured sensory scenes[1]. However, two central characteristics of cortical circuits remain to be properly integrated into population coding frameworks. First, neuronal activity in sensory cortices is often noisy, showing significant variability of spiking responses evoked by the same stimulus[2,3]. In many traditional coding frameworks such spiking variability degrades the representation of stimuli by cortical activity[4]. Why cortical responses display large spiking variability while isolated cortical neurons can respond reliably remains far from clear. Second, the primary source of synaptic inputs to cortical neurons does not come from upstream centers which convey sensory signals, but rather from recurrent pathways between cortical neurons[5–7]. While such recurrent connections are often organized about a stimulus feature axis[8,9], it is not obvious whether or how their presence improves overall representation. We propose a biologically motivated inference coding scheme where these two ubiquitous cortical circuit features, variability in spike generation and recurrent connections, together support a probabilistic representation of stimuli in rich sensory scenes.

Numerous studies have framed sensory processing in the cortex in terms of Bayesian inference (e.g., refs. 10–16). Specifically, the 'Bayesian brain' hypothesis posits that sensory cortex infers and synthesizes a posterior distribution of the latent stimuli which describes the probability of possible stimuli that could have given rise to the sensory inputs. Performing Bayesian inference requires cortex to store an internal model that represents how sensory inputs and external

[1]Department of Neurobiology and Statistics, University of Chicago, Chicago, IL, USA. [2]Grossman Center for Quantitative Biology and Human Behavior, University of Chicago, Chicago, IL, USA. [3]Department of Mathematics, University of Pittsburgh, Pittsburgh, PA, USA. [4]Center for the Neural Basis of Cognition, Pittsburgh, PA, USA. [5]School of Psychological and Cognitive Sciences, Peking University, Beijing 100871, China. [6]IDG/McGovern Institute for Brain Research, Peking University, Beijing 100871, China. [7]Peking-Tsinghua Center for Life Sciences, Peking University, Beijing 100871, China. [8]Center of Quantitative Biology, Peking University, Beijing 100871, China. [9]Department of Mathematics, University of Houston, Houston, TX, USA. [10]Department of Biology and Biochemistry, University of Houston, Houston, TX, USA. [11]Present address: Lyda Hill Department of Bioinformatics, UT Southwestern Medical Center, Dallas, TX, USA. [12]These authors contributed equally: Krešimir Josić, Brent Doiron. ✉e-mail: kresimir.josic@gmail.com; bdoiron@uchicago.edu

stimuli are generated. Once a sensory input is received, cortical dynamics inverts this internal model in a process termed "analysis-by-synthesis"[12], and represents the posterior distributively across neurons and/or across time[15,16]. In this study, we propose that recurrent connections in cortical circuits store the prior of latent stimuli to produce the posterior distribution when combined with evidence from sensory inputs. Moreover, we posit that Poisson spiking variability provides a source of fluctuations needed for generating random samples from the inferred posterior.

To test these hypotheses, we consider a recurrent circuit model where neurons receive stochastic feedforward inputs which carry information about the external world, and respond with Poisson-distributed spiking activity. We find that such Poissonian spiking provides the variability that allows the network to generate samples from posterior stimulus distributions with differing uncertainties. We use this sampling framework to illustrate circuit-based Bayesian inference given two distinct generative models of stimuli in the external world: one organized hierarchically with a stimulus variable that depends on a latent stimulus parameter, and a second where a pair of latent stimuli are organized in parallel. In both cases, a recurrent circuit is able to generate samples from the joint posterior, and infer the values of the latent variables. We show through both analytic derivation and simulations that recurrent connections represent the correlation structure of these models, and the weight of these connections can be tuned to optimally capture the prior distribution of stimuli in the external world. The stronger the correlation between the latent variables, the stronger the recurrent connections need to be for the network to generate samples from the correct posterior distribution.

Finally, a neural signature of this circuit-based sampling mechanism is internally generated population noise correlations aligned with the stimulus response direction, often referred to as "differential correlations"[4,17]. In our framework, the amplitude of internally generated differential correlations is determined by the recurrent connection strength, which also determines the prior stored by the circuit. Since optimal inference requires a specific magnitude of recurrent connectivity, differential correlations resulting from such recurrent connectivity are a potential signature of optimal coding. This is in contrast to the deleterious impact of externally generated differential correlations. We thus predict that the correlation structure of the external world shapes recurrent wiring in neural circuits, and is reflected in the pattern of differential noise correlations. We use this logic to provide testable predictions from our framework for sampling-based Bayesian inference by recurrent, stochastic cortical circuits.

## Results

### Recurrent circuitry and spiking variability do not improve conventional neural codes

We start with the classic example of a sensory stimulus, $s$, encoded in neuronal population activity, $\mathbf{r}$, from which a stimulus estimate $\hat{s}$ can be decoded (Fig. 1a, top)[18]. It is reasonable to expect that neuronal circuitry is adapted to accurately represent ethologically relevant stimuli. However, as we will show next, in simple coding schemes two ubiquitous features of cortical circuits – internal spiking variability and recurrent connectivity – are at best irrelevant for, and in many cases degrade, the accuracy of these representations.

In population coding frameworks stimuli are encoded by a neuronal population with individual neurons tuned to a preferred stimulus value. The preferred values of all neurons cover the whole range of stimuli[18–20] (Fig. 1b, bottom); if $s$ ranges over a periodic domain (such as the orientation of a bar in a visual scene, or the direction of an arm reach) then it is commonly assumed that the neurons' preferred stimuli are distributed on a ring (Fig. 1b, top). To generate neuronal responses from such a population we simulate a network of neurons whose spiking activity, $\mathbf{r}_t$, at time $t$ is Poissonian with instantaneous firing rate $\boldsymbol{\lambda}_t$ (Eq. (11)). For simplicity we assume linear (or linearized) neuronal transfer

and synaptic interactions (Eqs. (10), (11)), so that the firing rate is a linear function of the feedforward and recurrent inputs. We couple excitatory (E) neurons with similar stimulus preferences more strongly[8,9] to one another, compared to neuron pairs with dissimilar tuning. In this way, the recurrent E connectivity has the same circular symmetry as the stimulus (Fig. 1b). In contrast, connections between inhibitory (I) neurons are unstructured, and inhibitory activity acts to stabilize network activity[21]. A stimulus, e.g., $s = 0$, results in elevated activity of E neurons with the corresponding preference (Fig. S1a). As expected, an increase in the strength of recurrent excitatory connections increases both the firing rates and the trial-to-trial pairwise covariability (i.e., noise correlations) in the responses[2] (Fig. S2a). This canonical network model has been widely used to explain cortical network dynamics and neural coding[21–23]. And our network model can produce neuronal responses that are qualitatively similar to experimental observations, including the variance of neuronal firing rate, the Fano factor, and the noise correlations (Fig. S2b–d).

We use linear Fisher Information (LFI) to quantify the impact of recurrent connectivity and internal spiking variability on the accuracy of the stimulus estimate, $\hat{s}_t$, from the activity vector $\mathbf{r}_t$ (see details in Eq. S39 in Supplementary Information). The inverse of LFI provides a lower bound on the expected square of the difference between the true value, $s$, and the estimate, $\hat{s}_t$, made by a linear decoder[1,4,17–19,24]. In the limit of an infinite number of neurons available to the decoder LFI is unaffected by recurrent connectivity strength, $w_E$ (Fig. 1d, dashed line). This is because the mean response of the network is linear in its inputs, and an (invertible) linear transformation can neither increase nor decrease LFI (see Eq. S38 in Supplementary Information). For networks with a finite number of neurons, the variability from spike generation is shared between neurons via recurrent interactions. Consequently, an increase in coupling strength, $w_E$, reduces LFI in finite networks (Fig. 1d, colored lines).

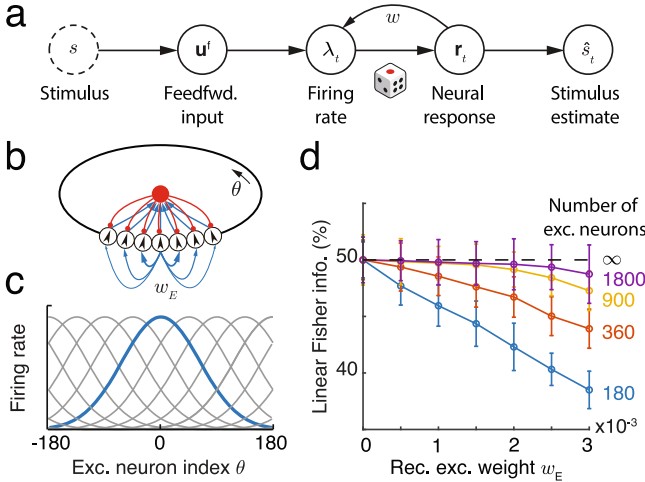

**Fig. 1 | A network with structured recurrent connections limits the linear Fisher Information (LFI) about external stimuli. a** A schematic diagram showing how a stimulus, $s$, is encoded in neuronal response, $\mathbf{r}_t$. A stimulus estimate, $\hat{s}_t$, can be obtained from $\mathbf{r}_t$.. **b** A recurrent ring model (top) where the connections between excitatory neurons are dependent on their distance along the ring. Blue arrows: excitatory synapses with line width denoting connection strength; red arrows: inhibitory synapses. **c** The population activity of excitatory neurons in the ring model, $\mathbf{r}_t$, dependent on a stimulus, $s$. The blue curve shows the population activity in response to $s = 0$, and gray curves the activities in response to stimuli with values at the peak locations of the curves. **d** For finite size networks (colored lines; ratio of excitatory to inhibitory neurons kept constant) LFI decreases as $w_E$ increases. In the limit of infinite network size LFI does not depend on $w_E$ (dashed line). Since neural responses are variable, LFI in the neuronal response converges to only half of the LFI in the feedforward input. Error bars denote one standard deviation (SD), which were estimated from $N = 50$ independent samples generated by using Bootstrap.

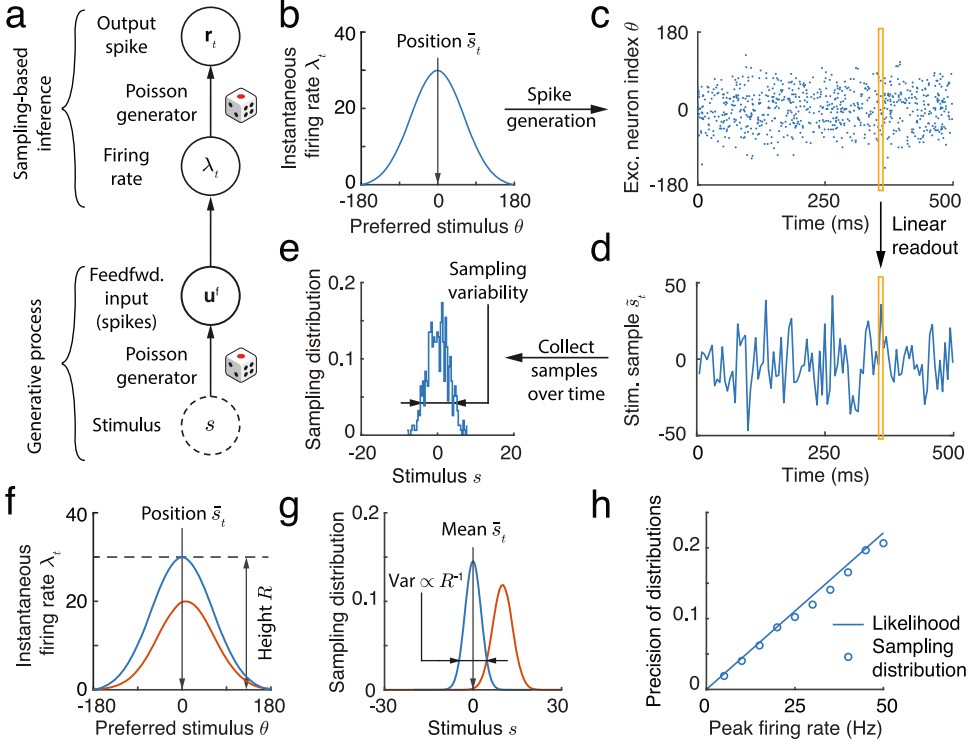

**Fig. 2 | Spike generation with Poissonian variability can support sampling-based Bayesian inference. a** We use a feedforward network model (no recurrent connections) to demonstrate how spiking variability drives sampling. Neurons receive feedforward inputs, $\mathbf{u}^f$, modeled as independent Poisson spike trains, resulting in a Poissonian population response, $\mathbf{r}_t$, with means determined by the instantaneous firing rate vector, $\boldsymbol{\lambda}_t$. (**b**–**e**) Demonstration of sampling via stochastic spike generation. A population of neurons with Gaussian tuning and firing rates $\boldsymbol{\lambda}_t$ (**b**) generates a realization of a population response, $\mathbf{r}_t$ (**c**). A sample from the posterior distribution of the stimulus (**d**, orange box) can be linearly read out from the population response (**c**, orange box). **e** The sampling distribution is obtained by collecting stimulus samples over time. The profile of population firing rates (**f**) determines the sampling distribution (**g**). The position of the population firing rate, $\bar{s}_t$, determines the mean of the sampling distribution, and the variance of the sampling distribution is inversely proportional to the peak firing rate, $R$. We show two population activity profiles, one in blue and the other in orange, to illustrate these points. **h** In an E-I network, the precision of the sampling distribution (the inverse of sampling variability) read out from E neurons increases with the height of firing rate, and is consistent with the likelihood directly read out from the feedforward input.

In sum, recurrent connectivity and spiking variability do not improve, and often degrade, stimulus representation in the network (as measured by LFI). Since synaptic coupling is biologically expensive, a network that most accurately and cheaply represents a stimulus is then one with no recurrent connections (i.e., $w_E = 0$) and minimal spiking variability. Nevertheless, connectivity in mammalian cortex is highly recurrent[5–7,9], and neural responses are highly variable[2,3]. What is then the function of these extensive recurrent connections between cortical neurons in information representation, and why are their responses so noisy?

While classical population code theory often explains how to generate point estimates of a stimulus (Fig. 1a), numerous studies suggest that the brain performs Bayesian inference to synthesize and estimate the probability distribution of latent stimuli from sensory inputs (e.g., refs. 10–15,25,26). To compute this posterior a neural circuit needs to combine a stored representation of the prior distribution of the stimulus with the likelihood conveyed by feedforward inputs. We propose that recurrent connectivity can be used to represent the prior and spiking variability can generate samples from this posterior distribution. Before we present our full model we first show how sampling-based inference can be implemented in a population of spiking neurons.

### Internally generated Poisson spiking variability drives sampling-based Bayesian inference

Many studies suggest that neuronal response variability is a signature of sampling-based Bayesian inference in neural circuits (e.g.,

refs. 16,27–34). In these studies, the instantaneous population responses, $\mathbf{r}_t$, represent a sample of a latent stimulus, and the empirical distribution of stimulus samples collected over time is an approximation of the posterior distribution. Implementing sampling requires a network that generates variable output with stable statistics. It has been well documented that cortical spiking responses are often approximately Poissonian[3,35]. Theoretical studies suggest that such Poissonian variability can be internally generated in a network with dynamically balanced recurrent excitation and inhibition[36,37]. We thus assumed that our model neurons are Poissonian, and used the resulting fluctuations as the internal source of variability needed for sampling-based Bayesian inference. It remains to be shown if discrete Poissonian variability can be used to generate samples from stimuli with continuous probability distributions (e.g., orientation, moving direction) with the flexibility needed to represent different stimulus uncertainties. However, spike counts are discrete, and it is possible that errors that arise from representing continuous parameters by discrete random variable are characteristic of stimulus inference by animals that use sampling.

We address this question using a theory based on a simple model network composed of excitatory (E) Poissonian neurons (Eqs. (10), (11)), and subsequently support our findings by simulating a network containing both E and inhibitory (I) neurons (e.g., Fig. 1b). We start by showing that Poissonian spiking in a population of tuned neurons can drive sampling from a well–defined distribution. We assume that the instantaneous firing rates of a population of E neurons, $\boldsymbol{\lambda}_t$, have a bell-shaped (Gaussian) profile (Fig. 2b), so that for the $j$th neuron

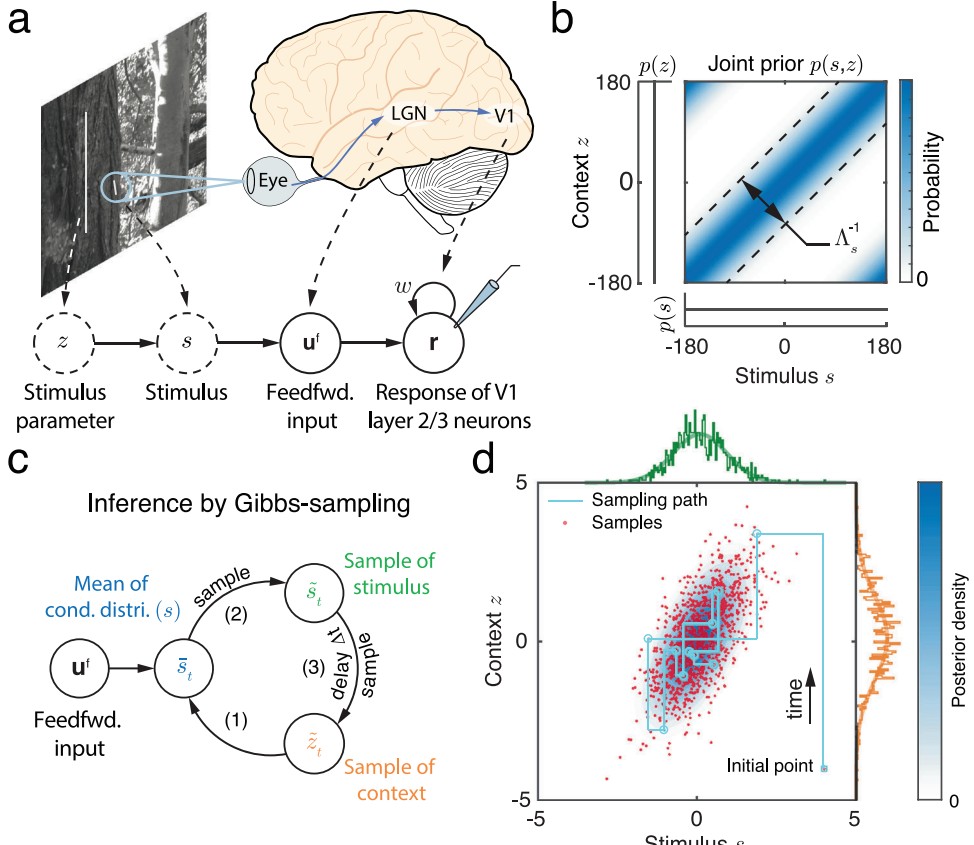

**Fig. 3 | A hierarchical generative model and posterior inference via Gibbs sampling. a** An example of sensory feedforward input generation: The stimulus parameter, $z$, is the orientation of the tree trunk, and the stimulus, $s$, is the orientation of the bark texture located in the classical receptive field of a V1 hypercolumn. The recurrent circuit generates samples from the joint posterior over stimulus and stimulus parameter. Solid circles: observations and responses in the brain; dashed circles: latent variables in the external world. Nature image is adapted from Tkačik, G. et al. Natural images from the birthplace of the human eye. PLoS one 6, e20409 (2011). **b** The joint prior over the stimulus parameter, $z$, and stimulus, $s$, is concentrated on the diagonal. The correlation between context and stimulus is determined by parameter $\Lambda_s$. **(c)** The posterior over stimulus parameter and stimulus can be approximated via Gibbs sampling (Eqs. (4a), (4c)) by iteratively generating samples of $s$ and $z$ from their respective conditional distributions. **d** The resulting approximations of the joint and marginal posterior over $s$ and $z$. Light blue contour: the posterior distribution (Eq. (24)); Red dots: Samples obtained using Gibbs sampling. The green and orange projections are the marginal posterior distributions of $s$ and $z$, respectively.

$\lambda_{tj} = R \exp[\mathbf{h}_j(\bar{s}_t)] = R \exp[-(\bar{s}_t - \theta_j)^2/2a^2]$ (See Eq. (12) in Methods). Here $\theta_j$ is the preferred stimulus of neuron $j$, $a$ is the width of the tuning curve, and $\bar{s}_t$ is the location of the peak of the firing rate profile, $\lambda_t$, in stimulus space (x-axis in Fig. 2b). Note that the value of $\bar{s}_t$ is arbitrary here, but we will later relate it to the input to the population. Finally, the preferred stimuli of the E neurons, $\{\theta_j\}_{j=1}^{N_E}$, are uniformly distributed over the stimulus range (Fig. 1b). In each time interval the population activity is given by a vector of independent Poisson random variables, $\mathbf{r}_t$, with means determined by the instantaneous firing rate vector $\lambda_t$ (Fig. 2b, c). At each time, $t$, this spiking activity produces a stimulus sample, $\tilde{s}_t$, from the probability distribution determined by the instantaneous firing rates, $\lambda_t$ (Fig. 2d, see Methods),

$$\tilde{s}_t \sim p(\tilde{s}|\lambda_t) \propto \exp[\mathbf{h}(\tilde{s})^\top \lambda_t] \propto \mathcal{N}(\tilde{s}|\bar{s}_t, \Lambda^{-1}). \quad (1)$$

With the Gaussian firing rate profile we use here, the stimulus sample, $\tilde{s}_t$, can be read out as $\tilde{s}_t = \sum_j \mathbf{r}_{tj}\theta_j / \sum_j \mathbf{r}_{tj}$ (Eq. (14) and Fig. 2d), which can be thought of as the location of the response, $\mathbf{r}_t$, in stimulus space (y-axis in Fig. 2c). The collection of stimulus samples across time ($\{\tilde{s}_t\}$; Fig. 2e), determines the sampling distribution $q(s) = T^{-1}\sum_t \delta(s - \tilde{s}_t)$ which approximates the distribution $p(s|\lambda_t)$, i.e., $p(s|\lambda_t) \approx q(s)$[16,38]. Here, $\delta(\cdot)$ is the Dirac delta function and $T$ is the number of samples. We assumed that instantaneous population firing rates are smooth to simplify the analysis, but this assumption is not essential. Sampling driven by Poissonian variability will work as long as the temporally averaged population firing rate is smooth, even if the instantaneous population firing rate is noisy (see Eq. (17)).

To use this mechanism to produce samples from the posterior distribution of a stimulus, we must define a generative model for the feedforward inputs evoked by a stimulus. We take the feedforward input to the neural population, $\mathbf{u}^f$, to be a vector of independent Poisson spike counts with Gaussian tuning over the stimulus, $s$. Following assumptions widely used in previous studies of probabilistic population codes (PPC)[39,40], we assume that the mean input spike count to the $j$th excitatory neuron in the population is $\langle \mathbf{u}_j^f(s) \rangle \propto \exp[\mathbf{h}_j(s)] = \exp[-(s - \theta_j)^2/2a^2]$. A single realization of the input, $\mathbf{u}^f$, in a time interval encodes the whole likelihood function over the stimulus, $p(\mathbf{u}^f|s)$[39]. This likelihood is proportional to a Gaussian due to the Gaussian profile of feedforward input (Eq. (19)),

$$\begin{aligned} p(\mathbf{u}^f|s) &= \prod_{j=1}^{N_E} \text{Poisson}\left[\langle \mathbf{u}_j^f(s) \rangle\right], \\ &\propto \exp\left[\mathbf{h}(s)^\top \mathbf{u}^f\right], \\ &\propto \mathcal{N}(s|\mu_f, \Lambda_f^{-1}). \end{aligned} \quad (2)$$

Here the likelihood mean, $\mu_f$, is determined by the location of $\mathbf{u}^f$ in stimulus space, and the precision, $\Lambda_f$, is proportional to the spike count

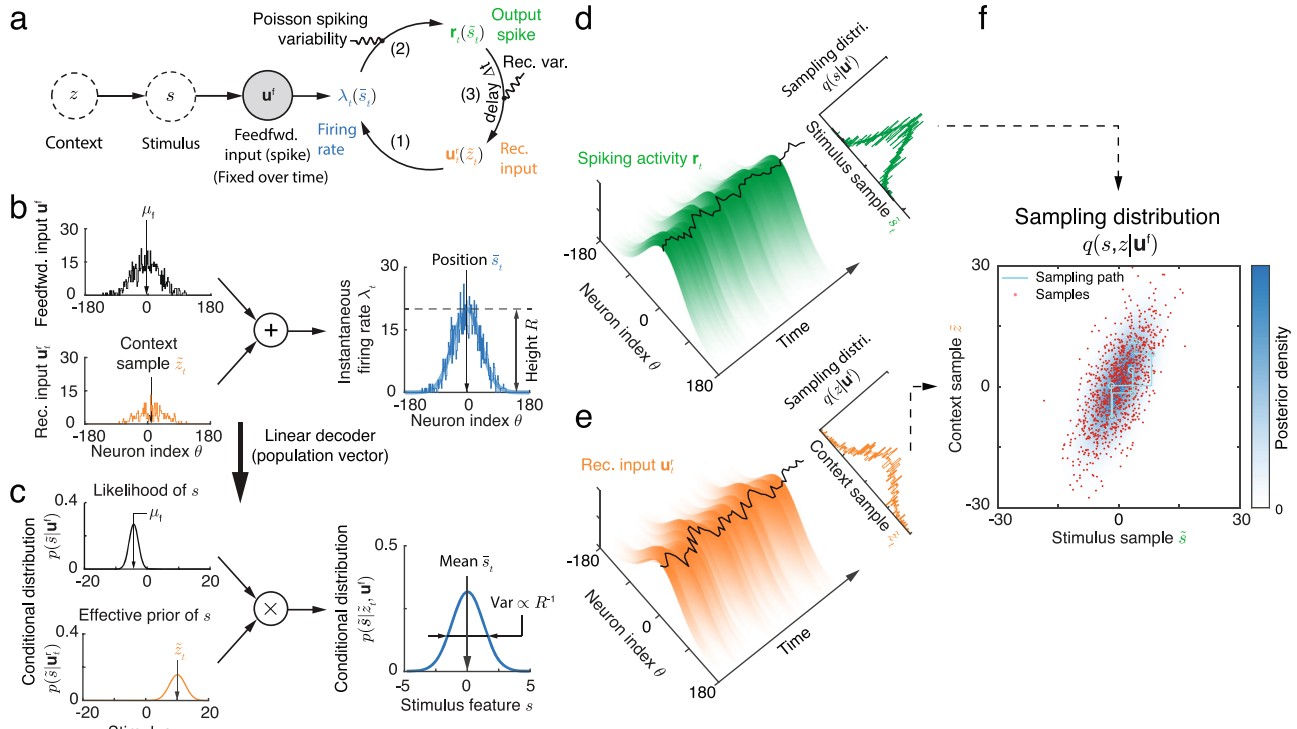

**Fig. 4 | A recurrent circuit generates samples from the posterior defined by a hierarchical generative model. a** Schematic of recurrent circuit dynamics, in which stimulus, $s$, and stimulus parameter, $z$, are encoded respectively in the population response, $\mathbf{r}_t$, and recurrent inputs, $\mathbf{u}_t^r$. **b, c** When the feedforward inputs and recurrent inputs share the same tuning profile, summing the two inputs to define the instantaneous firing rate (**b**) is equivalent to multiplying the conditional distributions encoded by the two inputs to obtain the conditional distribution of the stimulus, $p(s|\tilde{z}_t, \mathbf{u}^f)$. **c** The conditional distributions of the stimulus can be explicitly read out from corresponding population responses by a linear decoder (**b**). **d**–**f** Reading out the joint sampling distribution from the recurrent circuit. The

projection of the spiking activity (Eq. (14)) and recurrent inputs (Eq. (29)) onto the stimulus subspace (black curves), can be read out linearly from the population activity and interpreted as a sample of stimulus and stimulus parameter respectively (Eqs. (4b), (4c)). Top right insets: the empirical marginal distributions of samples and marginal posteriors (smooth lines). (**f**) The joint value (red dots) of instantaneous samples of stimulus (black curve on the surface in (**d**)), and stimulus parameter (black curve on the surface in (**e**)) represent samples from the joint posterior of the stimulus and stimulus parameter. The true joint posterior is represented by the blue contour.

(or height) of $\mathbf{u}^f$ (Eq. (20)). Since a realization of the feedforward input encodes the whole likelihood function, we present a fixed $\mathbf{u}^f$ to the network over time (dropping the time index $t$), and describe how samples from the posterior $p(s|\mathbf{u}^f)$ are generated by the network.

A simple example of inference via sampling is provided by a population of E neurons without recurrent connections and instantaneous firing rates equal to the feedforward input, $\boldsymbol{\lambda}_t = \mathbf{u}^f$ (Eq. (10)), and hence constant in time (Fig. 2a). In this feedforward network Poisson spike generation produces samples from the normalized likelihood, i.e., $\tilde{s}_t \sim p(\tilde{s}|\boldsymbol{\lambda}_t) \propto p(\mathbf{u}^f|\tilde{s})$, and consequently the network represents a uniform stimulus prior (i.e., $p(s)$ is a constant).

To test our theory, we simulated the response of a network of tuned excitatory (E) and untuned inhibitory (I) neurons (Fig. 2a, c) to a fixed but randomly generated feedforward input (Eq. (18)). While the E neurons shared no recurrent connections, the E and I neurons were connected to maintain stable network activity. To confirm that the overall firing rate dictated the sampling variability (Eq. (1)), we increased the feedforward input rate, which reduced the width of the likelihood (Eq. (2)). As a result, the sampling precision (inverse of the sampling variance) increased and matched the precision of the likelihood (Fig. 2g, h), even as the normalized response variability (measured the by Fano factor) of single neurons remained unchanged.

While the above analysis introduces the key components of a sampling-based theory of inference, stimulus sampling using a feedforward network is unnecessary: A single observation of the response $\mathbf{r}$ in a deterministic feedforward network ($\mathbf{r} = \mathbf{u}^f$ after removing spike generation in Eq. (11)) would also represent the whole likelihood[39],

avoiding the costly process of collecting samples $\tilde{s}_t$ across time. We next consider more interesting cases, and show that spiking variability in recurrent networks can drive sampling from more complex posterior distributions.

## Recurrent cortical circuit samples a hierarchical generative model

Recurrent networks can store a variety of generative model structures; to demonstrate the generality of our sampling framework we provide two example generative models which serve as building blocks for more complex models. We first consider a two-stage hierarchical model of feedforward inputs received by the cortical circuit (Fig. 3a). The first stage of our model consists of a stimulus, $s$, and a stimulus parameter, $z$, both of which are one dimensional for simplicity. The structure of the world is described by the joint distribution, $p(s, z)$. Using the visual system as motivation, $s$, could be the orientation of the visual texture within a classical receptive field (local information) of a hypercolumn of V1 neurons, while stimulus parameter, $z$, may refer to the context orientation within a non-classical receptive field of these cells (Fig. 3a). The likelihood of the stimulus based on a given parameter, $p(s|z) = \mathcal{N}(s|z, \Lambda_s^{-1})$, is Gaussian with precision $\Lambda_s$. For simplicity, we assume that the prior, $p(z)$, is uniform, which implies that the marginal prior of $s$, is also uniform (Fig. 3b). This assumption is not essential for our main conclusions but does simplify the analysis. Importantly, the joint prior of stimulus and stimulus parameter, $p(s, z)$, can have non-trivial structure with the density concentrated around the diagonal $s = z$ (Fig. 3b). The

precision $\Lambda_s$ measures how strongly $z$ and $s$ are related, and thus determines how strongly their joint distribution is concentrated around the diagonal.

The second stage of the generative model describes how the feedforward input depends on the stimulus, $s$; this is identical to our prior treatment (See Eq. (2)). Combining these two stages provides a complete description of the generative model for the feedforward input received by neurons in the population,

$$p(\mathbf{u}^{\mathrm{f}}|s)p(s|z)p(z) \propto \prod_{j=1}^{N_E} \mathrm{Poisson}\left(\mathbf{u}_j^{\mathrm{f}}|s\right)p(s|z),$$
$$\propto \mathcal{N}(s|\mu_{\mathrm{f}},\Lambda_{\mathrm{f}}^{-1})\mathcal{N}(s|z,\Lambda_s^{-1}). \quad (3)$$

Given this hierarchical model, we can show that the joint posterior over stimulus and stimulus parameters, $p(s,z|\mathbf{u}^{\mathrm{f}})$ is a bivariate normal distribution (see Eq. (24)), and we next use it to evaluate the accuracy of the sampling distribution.

**Gibbs sampling of the joint posterior of stimulus and stimulus parameter.** One approach to approximate the joint distribution over stimulus and stimulus parameter is Gibbs sampling[31,38,41,42] which starts with an initial guess for the value of the two latent variables, and proceeds by alternately generating samples of one variable from the distribution conditioned on the value of the second variable. More precisely, to approximate the joint posterior of $s$ and $z$ (Eq. (3)), Gibbs sampling proceeds by generating a sequence of samples, $(\tilde{s}_t,\tilde{z}_t)$ indexed by time $t$, through recursive iteration of the following steps (Fig. 3c and Eq. (25)),

$$\mathrm{Compute}: p(\tilde{s}|\tilde{z}_t,\mathbf{u}^{\mathrm{f}}) \propto p(\mathbf{u}^{\mathrm{f}}|\tilde{s})p(\tilde{s}|\tilde{z}_t) \equiv \mathcal{N}(\tilde{s}|\bar{s}_t,\Lambda^{-1}), \quad (4a)$$

$$\mathrm{Sample}: \tilde{s}_t \sim p(\tilde{s}|\tilde{z}_t,\mathbf{u}^{\mathrm{f}}), \quad (4b)$$

$$\mathrm{Sample}: \tilde{z}_{t+\Delta t} \sim p(\tilde{z}|\bar{s}_t) = \mathcal{N}(\tilde{z}|\bar{s}_t,\Lambda_s^{-1}). \quad (4c)$$

Here $\Delta t$ is the time increment between successive samples. The samples (red dots in Fig. 3d) are generated by alternately fixing the values of the two variables, so that sampling trajectories alternate between horizontal and vertical jumps (cyan lines in Fig. 3d). The empirical distribution of samples, i.e., $q(s,z|\mathbf{u}^{\mathrm{f}}) = T^{-1}\sum_t \delta[(s,z)^\top - (\tilde{s}_t,\tilde{z}_t)^\top]$ with $\top$ denoting vector transpose, approximates the joint posterior $p(s,z|\mathbf{u}^{\mathrm{f}})$ (blue contour map in Fig. 3d, Eq. (24))[38]. To approximate $p(s|\mathbf{u}^{\mathrm{f}})$, the marginal posterior distribution of $s$, we can use only samples $\tilde{s}_t$ to obtain the approximating distribution $q(s|\mathbf{u}^{\mathrm{f}})$ (compare the two green lines at the margin in Fig. 3d). The same is true for the marginal posterior over $z$.

**Implementing Gibbs sampling of stimulus and stimulus parameter in a recurrently coupled cortical circuit.** An implementation of Gibbs sampling in a recurrent E circuit can be intuitively understood by comparing the recurrent network dynamics (Fig. 4a) with the dynamics described by the Gibbs sampling algorithm (Fig. 3c). In the recurrent network a stimulus sample, $\tilde{s}_t$, is represented by the activity of E cells, $\mathbf{r}_t$, while a stimulus parameter sample, $\tilde{z}_t$, is represented by recurrent inputs, $\mathbf{u}_t^{\mathrm{r}}$. To generate correct samples we require that the conditional distribution that is represented by the instantaneous firing rate, $\lambda_t$ (Eq. (1)), matches the conditional distribution used in the Gibbs sampling algorithm (Eq. (4b)), so that $p(\tilde{s}|\tilde{z}_t,\mathbf{u}^{\mathrm{f}}) = p(\tilde{s}|\lambda_t) \propto \exp[\mathbf{h}(\tilde{s})^\top \lambda_t]$. Equating the two distributions (see Eqs. (4a) and (10)) yields the relation,

$$\ln p(\tilde{s}|\tilde{z}_t,\mathbf{u}^{\mathrm{f}}) = \ln p(\mathbf{u}^{\mathrm{f}}|\tilde{s}) + \ln p(\tilde{s}|\tilde{z}_t),$$
$$\iff \mathbf{h}(\tilde{s})^\top \lambda_t = \mathbf{h}(\tilde{s})^\top \mathbf{u}^{\mathrm{f}} + \mathbf{h}(\tilde{s})^\top \mathbf{u}_t^{\mathrm{r}}. \quad (5)$$

This equation holds when two constraints are satisfied: First, the firing rate vector, $\lambda_t$, needs to have a Gaussian profile peaked at $\bar{s}_t$, i.e., the mean of $p(\tilde{s}|\tilde{z}_t,\mathbf{u}^{\mathrm{f}})$ (Eq. (4a)). Second, the peak firing rate, $R$, needs to be proportional to the precision of $p(\tilde{s}|\tilde{z}_t,\mathbf{u}^{\mathrm{f}})$, i.e., $R \propto \Lambda$ (see Fig. 2f, g). In a neural circuit one way for $\lambda_t$ to satisfy these constraints is for feedforward inputs, $\mathbf{u}^{\mathrm{f}}$, and recurrent inputs, $\mathbf{u}_t^{\mathrm{r}}$, to both have Gaussian profiles with the same width, $a$, as that of $\lambda_t$ (by sharing the same $\mathbf{h}(\tilde{s})$, Eqs. (5) and (12)). This is because the sum of two Gaussian-profile inputs with the same width, $a$, gives a firing rate, $\lambda_t$, with the same tuning, as long as the difference of the locations of two inputs is much smaller than the width, $a$. Our generative model (Eq. (3)) produces feedforward input, $\mathbf{u}^{\mathrm{f}}$, with a Gaussian profile and encodes the likelihood function $p(\mathbf{u}^{\mathrm{f}}|\tilde{s})$. The recurrent input, $\mathbf{u}_t^{\mathrm{r}}$, then need to represent the conditional distribution $p(\tilde{s}|\tilde{z}_t)$. Hence, to satisfy Eq. (5) the recurrent input $\mathbf{u}_t^{\mathrm{r}}$ should have the same Gaussian profile as $\mathbf{u}^{\mathrm{f}}$ (Eq. (29)), with its location and magnitude determined by the mean and precision of $p(\tilde{s}|\tilde{z}_t)$, respectively.

If recurrent interactions are absent (setting $\mathbf{u}_t^{\mathrm{r}} = 0$), then network activity, $\mathbf{r}_t$, generates samples from the normalized likelihood, $p(\mathbf{u}^{\mathrm{f}}|\tilde{s})$, as we showed previously when describing feedforward networks (Fig. 2). When neurons only receive recurrent inputs (setting $\mathbf{u}^{\mathrm{f}} = 0$), the network generates samples from the conditional distribution $p(\tilde{s}|\tilde{z}_t)$. Driven by a sum of recurrent and feedforward inputs, the network generates samples from a distribution given by the product of the conditional distributions encoded by both inputs respectively (Fig. 4b, c).

The recurrent weights must be adjusted so that the recurrent input has the appropriate magnitude and width to encode the likelihood $p(s|z)$. To simplify the exposition we first assume that E neurons are only self-connected, so that the width of recurrent input trivially matches that of the feedforward input (otherwise recurrence will broaden the profile of the firing rate activity $\lambda_t$ over the network). To constrain the magnitude of the recurrent weights we require that the sum of the recurrent inputs satisfies $\sum_j \mathbf{u}_{tj}^{\mathrm{r}} \propto \Lambda_s$. Since $\mathbf{u}_j^{\mathrm{r}} = w_E \mathbf{r}_j$ and the width of $\mathbf{u}_j^{\mathrm{r}}$ and $\mathbf{r}_j$ are equal, the magnitude of the recurrent weights that result in samples from the correct posterior must satisfy:

$$w_E^* = \frac{\langle \mathbf{u}_j^{\mathrm{r}} \rangle}{\langle \mathbf{r}_j \rangle} = \frac{\langle \sum_j \mathbf{u}_j^{\mathrm{r}} \rangle}{\langle \sum_j \mathbf{r}_j \rangle} = \frac{\Lambda_s}{\Lambda_{\mathrm{f}} + \Lambda_s}, \quad (6)$$

where $\Lambda_s$ and $\Lambda_{\mathrm{f}}$ are the precision of likelihood $p(s|z)$ and $p(\mathbf{u}^{\mathrm{f}}|s)$ respectively (Eq. (3)). The optimal recurrent weight, $w_E^*$, thus encodes the correlation between the stimulus $s$ and the stimulus parameter $z$. An increase in correlation between $s$ and $z$, resulting in a narrower diagonal band in $p(s,z)$ (Fig. 3b), requires an increase in the recurrent weight $w_E^*$ for optimal sampling. When the underlying parameter and stimulus are uncorrelated so that $\Lambda_s = 0$, the hierarchical generative model (Fig. 3a) is equivalent to the generative model without stimulus parameter (Fig. 2a) and recurrent interactions are not needed for sampling (i.e., $w_E^* = 0$). Moreover, the optimal recurrent weight also depends on the likelihood precision $\Lambda_{\mathrm{f}}$ that is determined by the input spike count. Hence, the optimal weight needs to be adjusted depending on feedforward inputs so that samples from the correct posterior are generated (see Discussion of how this feature impacts the network sampling). Overall, our framework (Eq. (6)) thus predicts that optimal Bayesian inference is achieved with recurrent synaptic weights which depend on the correlative structure of the external world. We numerically test this prediction in the next section.

**A stochastic E-I spiking network jointly samples stimulus and stimulus parameter**
To confirm the predictions of this analysis, we simulated a full recurrent network consisting of both E and I neurons with Poisson spike train statistics (see details in Eqs. (47)–(50)). The E neurons were synaptically connected to each other (Eq. (49), see Fig. 1a), in contrast

to the simple network of self-connected E neurons we described above. While recurrent E to E coupling broadens the tuning of excitatory recurrent input, lateral inhibition can sharpen Gaussian firing rate profiles so that it matches that of the feedforward inputs (as required by Eq. (5)).

The activity of the recurrent network in response to a fixed but randomly generated feedforward input (Eq. (3)) can be decoded to produce samples from the bivarite posterior distribution of the stimulus and stimulus parameter. As above, samples from the conditional stimulus distribution are represented by the activity of E neurons (Eq. (14)), while samples from the conditional stimulus parameter distribution are represented by recurrent inputs received by E neurons (Eq. (29); black curves overlaid on the top of population responses in Fig. 4d, e, respectively). To update recurrent inputs we only used neuronal activity at the previous time step. Thus, the activities of E neurons and their recurrent inputs were updated in alternation, consistent with Gibbs sampling. The trajectory obtained by plotting the stimulus sample read out from the network activity on one axis, and plotting the stimulus parameter sample read out from recurrent E inputs on another axis then exhibits the characteristics of Gibbs sampling (Fig. 4f, cyan line). The resulting sampling distribution provides a good approximation to the joint posterior of stimulus and context (compare red dots and blue contour in Fig. 4f). Inhibitory neurons again did not respond selectively to either the stimulus or the stimulus parameter.

For the network to generate samples from the joint posterior, the recurrent connectivity should depend on the correlation between the stimulus and the stimulus parameter (Eq. (6)). To verify this prediction, we fixed the generative model (Eq. (3)) and changed only the recurrent weights in the network. For simplicity, we only varied the peak E weight, $w_E$ (Eq. (49)), and maintained network stability by fixing the ratio between E and I synaptic weights. While increasing $w_E$ did not change the sampling mean, it did increase the variance of the stimulus parameter sampling distribution, and increased the correlation between stimulus and stimulus parameter samples (Fig. 5a).

We use Kullback–Leibler (KL) divergence to measure the distance between the sampling distribution, $q(s, z|\mathbf{u}^f)$, and the true posterior, $p(s, z|\mathbf{u}^f)$ (Eq. (24)). The KL divergence quantifies the loss of mutual information, measured in bits, between the latent variables ($s$ and $z$) and the feedforward inputs, $\mathbf{u}^f$, when the true posterior, $p$, is approximated by the distribution, $q$ (Eq. (42))[38]. The mutual information loss in the network is minimized at a unique value of the recurrent weight, $w_E^*$, at which the sampling distribution, $q$, best matches the posterior, $p$ (Fig. 5b, black circle). To confirm that this optimal recurrent weight,

$w_E^*$, increases with the correlation in the prior (precision $\Lambda_s$, Eq. (6)), we numerically obtained the recurrent weight that minimizes the mutual information loss for each value of $\Lambda_s$ in the generative model. These results confirmed the predictions of our theory (Eq. (6), Fig. 5c): When $\Lambda_s = 0$, i.e., when stimulus parameter and stimulus are uncorrelated, a network with no interactions performs best ($w_E^* = 0$), while for small $\Lambda_s$ (relative to $\Lambda_f$) the optimal weight $w_E^*$ is positive and increases with $\Lambda_s$. In total, we have described a potential mechanism for a recurrent network of spiking neurons to perform sampling-based Bayesian inference.

### Generating samples from multi-dimensional posteriors with coupled neural circuits

To demonstrate the generality of the proposed neural code we next consider a world described by a broad, rather than deep (hierarchical) generative model. Information about each of two latent stimuli, $\mathbf{s} = (s_1, s_2)$, is relayed by corresponding feedforward inputs received by a neural circuit (Fig. 6a). We assume the prior is a bivariate Gaussian distribution (Fig. 6b), i.e., $p(\mathbf{s}) \propto \exp[-\Lambda_s(s_1 - s_2)^2/2] \equiv \mathcal{N}(s_1 - s_2, \Lambda_s^{-1})$, so that $\Lambda_s$ ($\Lambda_s \geq 0$) characterizes the correlation between $s_1$ and $s_2$. Furthermore, each stimulus, $s_m$, independently generates feedforward spiking inputs, $\mathbf{u}_m^f$, each of which is received by a separate network and produces responses $\mathbf{r}_m$ for $m = 1, 2$ (Fig. 6a). Thus, the full generative model of the input has the form,

$$p(\mathbf{u}^f|\mathbf{s})p(\mathbf{s}) = \left[\prod_{m=1}^{2} p(\mathbf{u}_m^f|s_m)\right] p(s_1, s_2),$$
$$\propto \left[\prod_{m=1}^{2} \mathcal{N}(s_m|\mu_{fm}, \Lambda_{fm}^{-1})\right] \mathcal{N}(s_1 - s_2, \Lambda_s^{-1}). \tag{7}$$

The likelihood $p(\mathbf{u}_m^f|s_m)$ is the same as that given previously (Eq. (2)), where the feedforward inputs, $\mathbf{u}_m^f$, are again described by conditionally independent Poisson spike counts with Gaussian tuning over stimulus $s_m$. As a concrete example, the two stimuli, $s_m$, could represent orientations of local edges falling in the central receptive fields of a V1 hypercolumn (Fig. 6a, bottom), with each V1 hypercolumn modeled by a network producing the response $\mathbf{r}_m$ (Fig. 6a, top). Then $\Lambda_s$ characterizes a priori tendency of the stimuli to share similar orientations, and determines how likely two local edges are to be part of a global line, as in the case of contour integration[43,44]. However, the generative model defined by Eq. (7) is quite general and has been also used to explain multisensory cue integration[10] and sensorimotor learning[13].

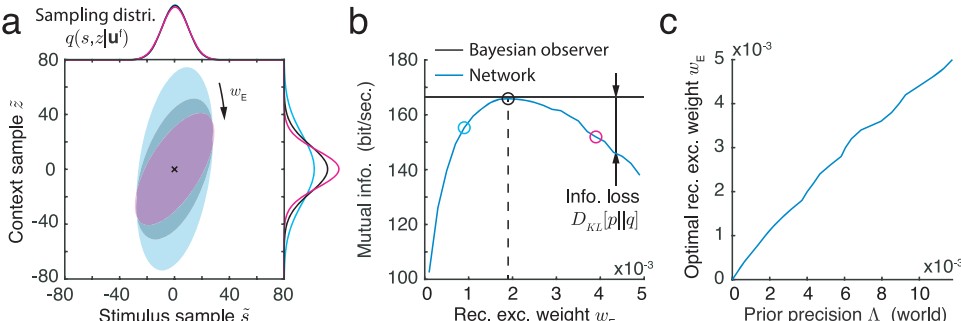

**Fig. 5 | The joint sampling distribution of stimulus and stimulus parameter changes with the recurrent weight in the network. a** The sampling distribution for different recurrent excitatory weights, $w_E$. The ratio of excitatory and inhibitory weights was fixed. Ellipses capture three standard deviations from the mean of the joint sampling distribution. Different colors correspond to the three values of $w_E$, denoted by different symbols in **b**. **b** The mutual information between the latent variables, $s$ and $z$, and the feedforward inputs for an ideal Bayesian observer (black horizontal line) and for the sampling distribution generated by the network model (blue curve). The difference between the two lines is the KL divergence between the posterior, $p(s, z|\mathbf{u}^f)$, and the sampling distribution, $q(s, z|\mathbf{u}^f)$. KL divergence is minimized when the weight in the recurrent network is set to a value, $w_E^*$, at which the sampling distribution, $q$, best matches the true posteriori, $p$ (black circle). **c** This optimal weight, $w_E^*$, increases with prior precision, $\Lambda_s$.

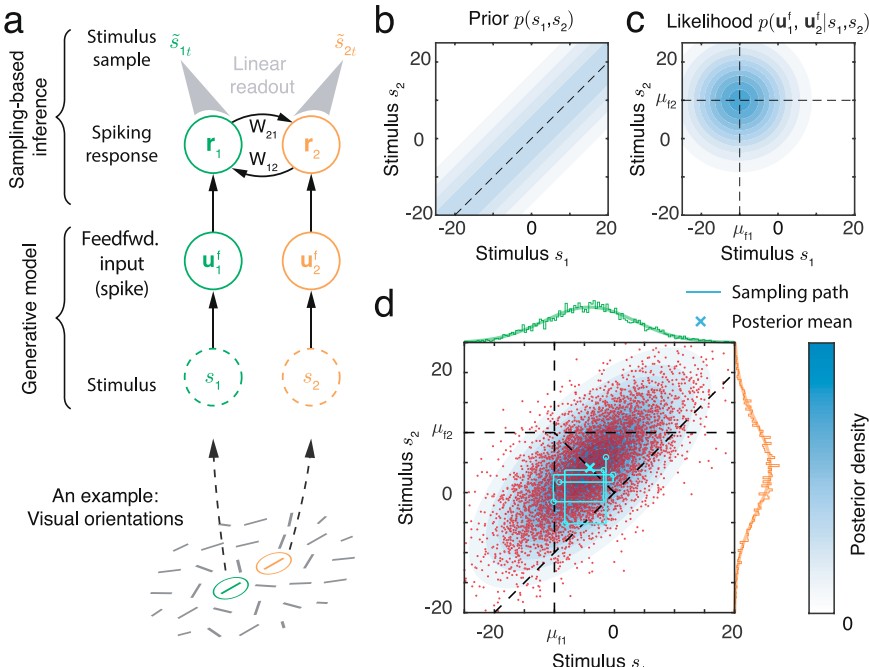

**Fig. 6 | Distributed sampling from a multivariate posterior distributions using coupled networks. a** Network $m$ ($m = 1, 2$) receives a feedforward input evoked by a stimulus, $s_m$. The coupling between the two networks represents the stimulus prior. A linear readout from each network, $m$, can be interpreted as a sample from the posterior of the stimulus, $s_m$. Examples of a prior (**b**) and likelihood (**c**). The prior distribution is concentrated around the diagonal line (dashed line), indicating the two stimuli are more likely to be colinear. In (**c**), $\mu_{f1} = -10$ and $\mu_{f2} = 10$ are the means of the likelihoods of $s_1$ and $s_2$, respectively. **d** The joint posterior of stimuli and the corresponding approximate sampling distribution generated by the coupled networks. A sample from the joint posterior can be read out individually from the activity of the corresponding network (shown in **a**). Light blue contour: the posterior distribution (Eq. (34)); Red dots: stimulus samples generated by the network.

The posterior is a bivariate Gaussian distribution (Fig. 6d, Eq. (34)) whose mean is shifted from the likelihood mean (Fig. 6c) towards to the diagonal line, because of the correlations between the stimuli in the prior (Fig. 6b). We can again use Gibbs sampling to approximate the posterior $p(\mathbf{s}|\mathbf{u}^f)$ using the following steps,

$$\text{Compute}: p(\tilde{s}_1|\mathbf{u}_1^f, \tilde{s}_{2,t-\Delta t}) \propto p(\mathbf{u}_1^f|\tilde{s}_1)p(\tilde{s}_{2,t-\Delta t}|\tilde{s}_1), \tag{8a}$$

$$\text{Sample}: \tilde{s}_{1t} \sim p(\tilde{s}_1|\mathbf{u}_1^f, \tilde{s}_{2,t-\Delta t}), \tag{8b}$$

where $\tilde{s}_{1t}$ and $\tilde{s}_{2t}$ are instantaneous samples at time $t$ of stimuli $s_1$ and $s_2$, respectively. We only give the steps needed to produce samples from the conditional distribution of $s_1$, as samples from the conditional distribution of $s_2$ can be obtained using the same steps after exchanging indices.

These sampling steps can be implemented distributively in a coupled neural circuit using a mechanism similar to that we described in the case of a hierarchical generative model. The activity of each network, $\mathbf{r}_m$, individually represents samples from the (marginal) posterior of $s_m$ (Fig. 6a, top). The joint posterior is then approximated as the collection of samples represented by the activity pairs $(\mathbf{r}_1, \mathbf{r}_2)$. Taking network $m = 1$ as an example, spike response $\mathbf{r}_{1t}$ produces a stimulus sample $\tilde{s}_{1t}$ as long as the instantaneous firing rate $\lambda_{1t}$ represents the conditional distribution $p(\tilde{s}_1|\mathbf{u}_1^f, \tilde{s}_{2,t-\Delta t})$ (Eq. (8a)). Since the feedforward input, $\mathbf{u}_1^f$, represents the likelihood $p(\mathbf{u}_1^f|\tilde{s}_1)$, to obtain the appropriate firing rates, $\boldsymbol{\lambda}_{1t}$, the recurrent input from network 2 to network 1, $\mathbf{u}_{12,t}^r$, must encode the correct conditional distribution, $p(\tilde{s}_{2,t-\Delta t}|\tilde{s}_1)$. As in the case of the mechanism we proposed to implement sampling as described by Eq. (5), $\mathbf{u}_{12,t}^r$ needs to have the same Gaussian profile as the firing rate $\boldsymbol{\lambda}_{1t}$, the position of $\mathbf{u}_{12,t}^r$ on the stimulus space should match the mean of $p(\tilde{s}_{2,t-\Delta t}|\tilde{s}_1)$, i.e.,

$\tilde{s}_{2,t-\Delta t}^* = \sum_j \mathbf{u}_{12,tj}^r \theta_j / \sum_j \mathbf{u}_{12,tj}^r$, and the magnitude of $\mathbf{u}_{12,t}^r$ must be proportional to the prior correlation, $\Lambda_s \propto \sum_j \mathbf{u}_{12,tj}^r$ (Eq. (39)). Hence, each network can sum the feedforward input and the recurrent input from its counterpart to obtain an update to the instantaneous conditional distribution given by Eq. (8a), and generate independent Poisson spikes to produce a sample from the instantaneous conditional distribution (Eq. (8b)). Notably, the sample of each stimulus can be locally read out from corresponding network (Eq. (41), Fig. 6a), even if the activities of two networks are correlated.

Since the recurrent input strength represents the stimulus correlation in the prior determined by precision $\Lambda_s$, the coupling between the two networks needs to be tuned to generate the appropriate recurrent input. Indeed, in a network with only E neurons, and connections only between neurons with the same preferred stimulus value but in different networks, the optimal homogeneous connection strength is $w_{mn}^* = \langle \mathbf{u}_{mn,j}^r \rangle / \langle \mathbf{r}_{n,j} \rangle = \Lambda_s / (\Lambda_{fn} + \Lambda_s)$ (Eq. (40)). This mirrors the result obtained with the hierarchical model presented earlier in Eq. (6).

**Coupled E-I spiking networks sample bivariate dimensional posteriors.** To test the feasibility of the proposed mechanisms for generating samples from a bivariate posterior, we simulated a pair of bidirectionally coupled circuits consisting of E and I neurons (Fig. 7a). This neural circuit model can be extended to generate samples from higher dimensional posterior distribution (see Discussion). Each circuit receives feedforward input generated by one of the two stimuli. On every time step the sample of each stimulus, $\tilde{s}_{mt}$, can be individually and linearly read out from the response of corresponding network, $\mathbf{r}_{mt}$ (Eq. (41)). Jointly, the two stimulus samples, one each from both networks, $\tilde{\mathbf{s}}_t = (\tilde{s}_{1t}, \tilde{s}_{2t})^\top$, provide a sample from the joint posterior of the two latent stimuli (Fig. 7b). We assumed that the synaptic connections between the networks, $w_{mn}$ ($m, n = 1, 2; m \neq n$), are excitatory, but

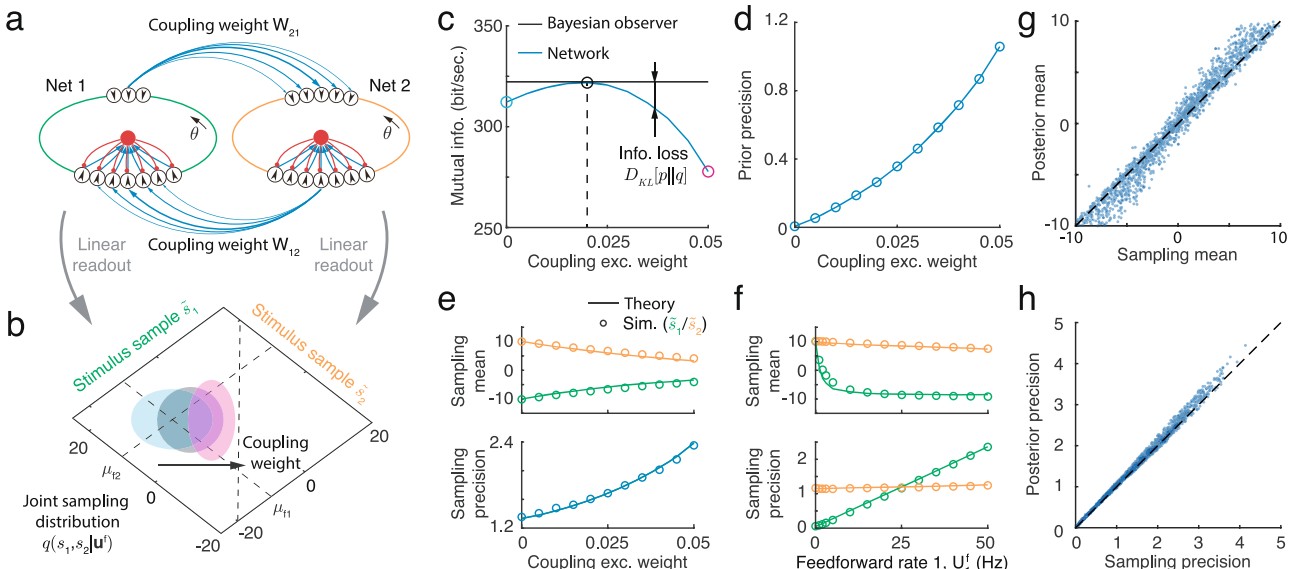

**Fig. 7 | The statistics of the multivariate sampling distribution of stimuli generated by coupled E-I circuits. a** Each of the two circuits individually generate a sample of a corresponding stimulus which can be read out linearly from that circuit's activity. Combining the readouts from the two networks yields the joint sampling distribution. The ring color indicates the stimulus sample the circuit generates: green and orange represent the stimulus $s_1$ and $s_2$, respectively. Blue arrows: E synapses with width denoting connection strength; red arrows: I synapses. **b** The sampling distribution shifts from the likelihood mean to the diagonal line as the coupling between the networks increases. Ellipses capture one standard deviation from the mean of the sampling distribution. Different colors correspond to the three different coupling weights between the circuits shown in (**c**). **c** The mutual information between latent variables and the feedforward inputs for the ideal Bayesian observer (black) and the sampling distributions generated by the

network with different coupling weights between the two circuits. **d** The optimal coupling weight that minimizes information loss also increases with prior precision (which is inversely proportional to the width of the band in Fig. 6b). **e** The mean and precision of the sampling distribution over the two stimuli change with the coupling weight between the circuits when the feedforward input is fixed. **f** The mean and precision of the sampling distribution over the two stimuli change with the firing rate of feedforward input to network 1, with other network parameters fixed. Comparison of the mean (**g**) and precision (**h**) of the sampling distributions with the posteriors under different combinations of feedforward inputs and coupling weights. Different dots are obtained from the sampling distributions obtained under different combinations of input direction and strength, and coupling weight between networks.

target both E and I neurons, while inhibitory connections are local to each network. We also adjusted network parameters so that the profiles of the inputs across networks (e.g., the inputs from network 2 to 1) have the same tuning profile as the feedforward inputs (see Methods). Since we assumed uniform marginal priors (see Eq. (32)), recurrent connections between E neurons within the a circuit were absent, while E and I neurons within a circuit were recurrently connected to ensure network stability. For simplicity, we chose parameters so that the two circuits were symmetric, but the strength of the feedforward inputs to each could differ.

We asked whether the activity of the two coupled circuits can generate samples from bivariate posteriors, and how the sampling distribution depends on the coupling, $w_{mn}$, between the two circuits. An increase in synaptic coupling between the two networks caused the sampling distribution to shift from the likelihood mean towards the diagonal (Fig. 7b), resulting in stimulus samples, $\tilde{s}_{1t}$ and $\tilde{s}_{2t}$ that were more similar. This is consistent with an increase in stimulus correlation in the multivariate prior, $\Lambda_s$ (Eq. (7)). To confirm our prediction that the optimal coupling strength between the two networks, $w^*_{mn}$, increases with the stimulus correlation in the prior, $\Lambda_s$, we numerically obtained the coupling weight that minimizes the loss of mutual information between latent stimuli and feedforward inputs (Fig. 7c). The optimal synaptic weight between the circuits increased with stimulus correlation in the prior. At the optimal weight, $w^*_{mn}$, the sampling distribution was close to the true posterior, showing that a properly tuned circuit can generate samples from the correct distribution (Fig. 7d).

We next asked how the sampling distribution in the network depends on network and feedforward input parameters. As the coupling between the two circuits increased, the sample means of both

stimuli converge (Fig. 7e, top) and the sampling precision of both stimuli increased as well (Fig. 7e, bottom), in agreement with a more correlated stimulus prior. We also tested whether a network with fixed parameters can generate samples from a family of posteriors with different uncertainties. To do so, we changed the uncertainty of the likelihood of $s_1$ by changing the firing rate in the feedforward input $\mathbf{u}^f_1$ received by network 1. We observed that with a narrower likelihood of $s_1$, the sample means of both stimuli shifted towards the mean of likelihood of $s_1$ (−10°), and sampling precision increased, consistent with a change in the posterior distribution (Fig. 7f). Lastly, to demonstrate the robustness of this network implementation of sampling-based inference we compare the sampling distributions to the true posteriors under different combinations of input and network parameters (Fig. 7g, h), in each case setting the recurrent coupling to the optimal value, $w^*_{mn}$, obtained numerically. Across different parameter values, we observe excellent agreement in both the mean (Fig. 7g) and precision (Fig. 7h) of the two densities. In sum, our recurrent network of spiking neuron models can be extended to support sampling-based Bayesian inference with multi-dimensional stimuli.

## A signature of stimulus sampling: internally generated differential noise correlations

A central prediction of our circuit framework for sampling-based Bayesian inference is that an increase in the correlation between stimuli in the sensory world should result in stronger synapses between neurons whose activities represent these stimuli (see Eq. (6)). This is a difficult prediction to test since measuring synaptic connectivity along a functional axis is already challenging[45], let alone measuring a change in synaptic strength owing to a change in stimulus statistics. Here, we outline a testable prediction of our theory by identifying a measurable,

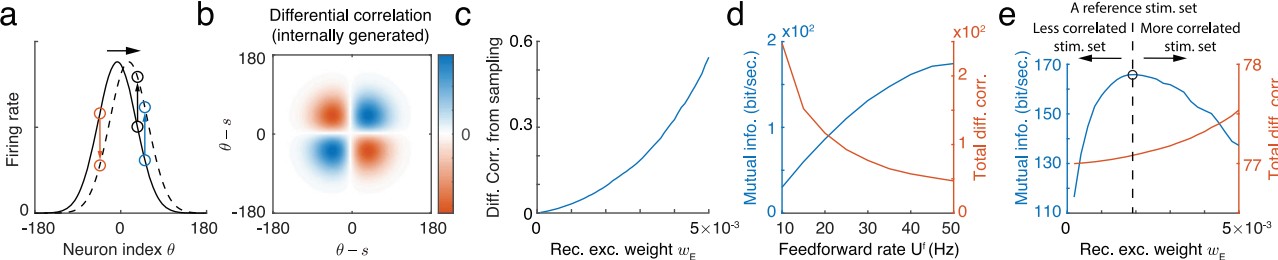

**Fig. 8 | Stimulus sampling by a network is reflected in the internally generated differential correlations, whose impact differs from differential correlations inherited from feedforward inputs. a** Stimulus sampling via spike generation causes the population firing rate to fluctuate along the stimulus subspace (x-axis). **b** The pattern of internally generated differential correlation in a network implementing sampling composed of neurons with Gaussian tuning. **c** Internally generated differential correlations in such a network increase with recurrent weight, $w_E$. **d** The rate in feedforward input decreases the externally generated correlations,

and increases the mutual information between the feedforward inputs and latent stimulus. **e** Recurrent network weights increase internally generated differential correlations. Mutual information between stimulus and feedforward inputs changes non-monotonically with recurrent weight. The direction of arrows indicates the predicted direction of change of the recurrent weights after an animal is retrained using a new stimulus set with different correlations compared to the reference stimulus set.

population-level signature of changes in functionally related recurrent synaptic strengths.

In response to a fixed feedforward input the responses of a recurrent circuit implementing stimulus sampling will fluctuate. The alignment of the recurrent circuitry and neuronal stimulus tuning causes a portion of these activity fluctuations to align with the subspace in which stimuli are coded. As an example, consider the sampling implemented by a single recurrent network (Fig. 4a), and suppose the population response fluctuates around its mean position (0° in the example of Fig. 8a), ignoring fluctuations along other directions in neuronal response space. The activity of neuron pairs with stimulus preference both above or below the mean position are positively correlated (the black and blue neurons in Fig. 8a), while the activity of neuron pairs with preferences straddling the mean are negatively correlated (the black and red neurons in Fig. 8a). Such stimulus sampling generates a covariance component which is proportional to the outer product of the derivative of neuronal tuning (Fig. 8b), i.e., $\mathbf{f}'_s \mathbf{f}'^{\top}_s$, where $\mathbf{f}'_s$ denotes the derivative of tuning $\mathbf{f}(s) = \langle \boldsymbol{\lambda}_t \rangle$ (mean firing rate) over stimulus $s$. Such noise correlations have been referred to as *differential correlations*[4,17], and are generally viewed as deleterious to stimulus coding. Stochastic sampling in coupled networks (Fig. 6a) produces similar differential noise correlations (see Supplementary Information).

In our network implementation of sampling, the amplitude of internally generated differential correlations is not arbitrary, but is determined by the recurrent connection strength, $w_E^*$. Here, the differential covariance matrix of population responses has the form (see Eq. (44))

$$\Sigma_{DC} = V(\bar{s}|\mathbf{u}^f)\mathbf{f}'_s \mathbf{f}'^{\top}_s,$$

$$\text{where } V(\bar{s}|\mathbf{u}^f) = \frac{\Lambda_s}{\Lambda_f(\Lambda_f + \Lambda_s)} = a^2 n_f^{-1} w_E^*, \tag{9}$$

where $V(\bar{s}|\mathbf{u}^f)$ is the variance of $\bar{s}_t$ in equilibrium over time, and $\bar{s}_t$ is the mean of the instantaneous conditional distribution (Eq. (4a)) represented by the position of instantaneous firing rate $\boldsymbol{\lambda}_t$ (Fig. 2b). Importantly, the amplitude of differential correlations increases with the recurrent weight, $w_E^*$, which is set by the prior precision $\Lambda_s$ (Eq. (6); Fig. 8c). Thus, in our framework internally generated differential correlations are a by-product of inference by sampling from posterior distributions of stimuli in a structured world.

**Distinguishing external and internal differential correlations.** The previous analysis of internally generated differential correlations in a circuit implementing sampling-based inference is based on the assumption of a fixed feedforward input (Eq. (9)). However, in typical

neurophysiology experiments an external stimulus, $s$, is fixed, while the feedforward input, $\mathbf{u}^f$, fluctuates due to variability in sensory acquisition and transmission noise (Eqs. (3) and (7)). Hence, differential correlations of neuronal population responses are a combination of correlations inherited from feedforward input[46], and correlations generated by recurrent network interactions that align with the population stimulus tuning[24]. When the feedforward input is described by a hierarchical generative model (Eq. (2)), the total magnitude of differential correlations in the evoked response is $a^2 n_f^{-1} w_E \mathbf{f}'_s \mathbf{f}'^{\top}_s + a^2 n_f^{-1} \mathbf{f}'_s \mathbf{f}'^{\top}_s$ (see Eq. (46)), where the second term reflects differential correlations inherited from the feedforward input (compare with Eq. (9)). Although the two sources of differential correlations are intertwined in the neuronal response, they impact the information content differently thus offering a potential way to distinguish between them in neural data.

Externally generated differential correlations decrease with feedforward input rate, which could be modulated by visual stimulus strength such as contrast (Fig. 8d, red curve). As a consequence, the mutual information (the information between feedforward inputs $\mathbf{u}^f$ and the latent variables, i.e., $s$ and $z$, sampled by recurrent network in Fig. 4a, Eq. (42)) increases with feedforward input intensity (Fig. 8d, blue curve). We, therefore, have a monotonic, decreasing relationship between externally generated differential correlations and mutual information. This is expected since such inherited correlations always impair information processing, as observed previously[4,17]. In contrast, an increase in recurrent weights, $w_E$, increases internally generated differential correlations, but results in a non-monotonic change in mutual information (Fig. 8b). Hence there is a non-monotonic relation between internally generated differential correlations and the mutual information between stimulus and feedforward inputs. In sum, the impact of external and internal differential correlations on stimulus coding can be distinguished by their respective monotonic and non-monotonic relation with the mutual information between stimulus and response.

## Discussion

We have presented a framework in which neuronal response variability and recurrent synaptic connections, two ubiquitous features of cortex, are jointly used to implement sampling-based Bayesian inference in neuronal circuit models. Combining mathematical analysis and network simulations, we established that stereotypical Poisson variability of discrete spike counts can drive flexible sampling from a family of continuous distributions. The sampling statistics are determined by the structure of recurrent coupling, which stores information about the stimulus prior, and feedforward inputs conveying the stimulus likelihood. Sampling-based inference is implemented in two steps: the

instantaneous firing rate, determined by the sum of feedforward and recurrent inputs, represents the instantaneous conditional distribution of latent stimulus, while Poissonian variability in spike generation is used to generate a random stimulus sample from this conditional distribution. We have shown how sampling can be implemented using biologically feasible mechanisms for three different generative models of increasing complexity. The simplest model includes one latent stimulus, while the more complex models include multiple latent stimuli organized hierarchically or in parallel. These three generative models form the basic building blocks of more complex models. Thus our ideas can be extended to a wide range of perceptual and cognitive processes[47].

The neural code we described shares some features with codes described in previous studies, including parametric representations in probabilistic population codes (PPCs)[15,39,40], and sampling-based codes (SBCs)[16,27–32]. In our framework, the conditional distributions of latent variables is represented by instantaneous firing rates which linearly encode the logarithms of these conditional distributions, and have a mathematical form that is similar to that used in past studies describing PPCs (e.g., Eq. (5)). Further, the posterior is represented by stimulus samples generated through a random process, a feature of all SBCs. Despite these similarities, there are fundamental differences between the neural code we described and previously proposed PPCs and SBCs.

PPCs are generally implemented in networks with no internally generated variability, with stochasticity inherited from the stimulus. In contrast, our proposed network is doubly stochastic: The Poisson variability in the feedforward input allows a single realization of the feedforward input to represent the whole stimulus likelihood[39], while internally generated Poisson variability drives stimulus sampling. Further, in PPCs the posterior is represented parametrically by a one-shot neuronal response, while in our proposed network the joint posterior is approximated by a sequence of samples, each obtained as a linear readout from the instantaneous neuronal responses. Although it takes time to collect sufficiently many samples to approximate the posterior well, an advantage of sampling codes compared to PPCs is that inference with multivariate posteriors can be implemented using linearly coupled subnetworks (Fig. 6), with the number of subnetworks determined by the dimension of the latent stimulus features. In contrast, to represent an $M$-dimensional multivariate posterior using PPCs requires $N^M$ neurons in a linear network ($N$ is the number of neurons in representing each dimension) so that the number of neurons increases exponentially with the latent stimulus dimension, $M$[16]. Alternatively, coupled networks with $NM$ neurons can be used, but require complex, nonlinear coupling between these networks[48,49].

Neurons emit a discrete number of spikes, but their responses often need to represent continuous quantities. Most studies of neural sampling implicitly rely on approximating Poissonian spike counts with Gaussian variables (e.g., refs. 29,31,51). However, this approximation does not work well when only a few spikes are emitted. Here, we showed that discrete Poisson spike generation can be used to generate samples from a posterior distribution of a continuous stimulus feature using a temporally averaged, smooth population firing rate profile. Thus, we have shown how a sample from a continuous variable can be generated even with only a few spikes from the neuronal population. Moreover, conventional SBCs are used to generate samples directly in a neural space whose dimension is given by the number of neurons in the population[16,27,28,30–34,50], where a neuronal response, $\mathbf{r}_t$, is interpreted directly as a sample from the (marginal) posterior of neuronal responses, $p(\mathbf{r})$. Hence the posterior mean is the temporally averaged population response, and the covariance of population responses is the posterior covariance. In contrast, our proposed network generates samples in a low dimensional stimulus subspace embedded in high dimensional neural activity space. The linear projection of network activity, $\mathbf{r}_t$, onto the stimulus subspace

represents a sample from the stimulus posterior, similar to a previous study[29]. A computational benefit of sampling in a low dimensional stimulus subspace is convergence speed, as the volume of the stimulus subspace is significantly smaller than that of the neural activity space. Indeed, in our examples sequences of samples generated by a single recurrent network (Fig. 4) and coupled networks (Fig. 6) can both converge to an equilibrium distribution in less than 20 ms, which is fast enough to complete inference on a behaviorally relevant time scale (Fig. S6). Furthermore, the multiplication of probability distributions of latent stimulus, which is central to Bayesian inference (e.g., cue combination, decision making, see review in ref. 15), can be implemented by summing the inputs to a neuronal population (Eq. (5)). This follows from the fact that the instantaneous population input (or firing rate) linearly encodes the logarithm of a probability distribution (Eqs. (1) and (5)). In contrast, producing samples in neural activity space using conventional SBCs requires nonlinear operations in neural circuits in order to multiply probability distributions (or histograms) of the samples[15].

A recent study demonstrated that an E-I recurrent network of rate-based neurons can be numerically optimized for sampling-based Bayesian inference[32]. In contrast, we used a theoretical approach to derive a network model of simplified spiking neurons, which implements sampling-based inference. This allowed us to explicitly describe the putative neural mechanisms needed for such sampling. Although the two studies use different generative models and neural representations, the network models in both studies share some common characteristics: ring structure, Poisson-like response variability, and tuning-dependent noise correlation (Fig. 1d). This implies that the seemingly different generative models and neural representations in the two studies reflect more general principles, as suggested in[51]. It will be interesting to extend our theoretical approach to dynamical spiking neurons to determine how the timescales of neuronal dynamics and neuronal oscillations impact inference in rich, dynamic sensory scenes (see below).

Differential noise correlations generated by recurrent network interactions are a signature of network sampling in our framework (Figs. 5c and 8c). This is in contrast to earlier studies where differential correlations were inherited from feedforward inputs[17,52]. While internally generated differential correlations could also emerge from a recurrent circuit which is not implementing inference[22,24,52–55] or implementing inference via other algorithms[56], in our framework, the relation between the magnitude of internally generated differential correlations, the posterior uncertainty, and the strength of the recurrent synaptic weight (Eq. (9)) provides a clear test which can be used to verify our proposed circuit mechanism of sampling-based inference. One possible experimental approach would modulate the functional recurrent strength by using a perceptual learning task. Specifically, after using a reference stimulus set with a prescribed correlation between latent stimuli to fully train an animal, we expect that recurrent synaptic weights will strengthen or weaken to improve inference (Fig. 8e, dashed line). This will result in a fixed value of differential noise correlations in the population response due to the recurrent circuitry. Re-training with a stimulus set that has more (less) correlated latent stimuli compared to the reference set will cause the recurrent weights to increase (decrease) (Fig. 8e, red line). When the reference stimulus set is again used to drive task behavior, then performance (as a proxy of mutual information) will decrease, regardless of whether differential correlations have increased or decreased compared to those resulting from the reference stimulus set (Fig. 8e, arrows). In brief, the non-monotonic relationship between differential noise correlations and the mutual information between stimulus and responses which support Bayesian inference offers a clear (and falsifiable) experimental prediction.

Implementing sampling-based inference in our proposed network requires that feedforward and recurrent inputs have the same tuning

profile over the stimulus (Eq. (5)). This assumption is supported by experiments in layers 4 and 2/3 in mouse V1[8]. Moreover, the recurrent connections in our network model are translation-invariant in the stimulus subspace, an assumption widely used in studies of continuous attractor networks (CAN)[22,54,57,58], and a recent network model implementing sampling[32]. Perfectly translation-invariant connections are not strictly required for a circuit to implement sampling, but this assumption allows us to simplify the mathematical analysis. Adding randomness in recurrent connectivity would increases the variance of sampling distributions. We could then adjust the overall recurrent weight (a scalar) so that the sampling distribution matches the posterior, with no need to fine-tune individual synaptic weights in the network model. In the past, CANs have been shown to achieve maximal likelihood estimation (point estimate) via template matching[15,58,59]. Here we have shown that a network with CAN-like structure and internally Poisson spiking variability is able to perform sampling-based Bayesian inference. In our network correlations in the stimulus prior are represented by the strength of recurrent synaptic activity, which implies that the (subjective) prior precision in the network increases with the feedforward input strength.

To maintain a fixed prior in the network recurrent weights need to decrease with increased feedforward input strength which encodes the likelihood precision, $\Lambda_f$ (Eq. (6)). Therefore, the (subjective) prior stored in the network with fixed recurrent weights may differ from the objective stimulus prior in the world ($\Lambda_s$ in Eqs. (3) and (7)) with feedforward inputs of different strengths. One possibility is that the proposed network model does not generate samples from each distinct posterior determined by a specific feedforward input, $p(\mathbf{s}|\mathbf{u}^f)$, but rather generates samples from the average sampling distribution over all possible feedforward inputs and hence matches the average posterior distribution $\mathbb{E}_{p(\mathbf{u}^f)}[p(\mathbf{s}|\mathbf{u}^f)] = \mathbb{E}_{p(\mathbf{u}^f)}[q(\mathbf{s}|\mathbf{u}^f)]$, where $\mathbb{E}_{p(\mathbf{u}^f)}[\cdot]$ denotes the average over the distribution $p(\mathbf{u}^f)$. Since the proposed recurrent circuit is general, this result may explain one source of inductive bias in cortical processing[60]. On the other hand, sampling correctly from each specific posterior could be achieved using different biophysical mechanisms that can modulate synaptic strengths and that we have not included in our model. For instance, short-term synaptic depression[61] or spike frequency adaptation[62] are gain control mechanisms that would allow the recurrent input strength (representing the prior correlation) to remain relatively fixed despite an increase in the feedforward input strength. Another possibility is that the recurrent circuit represents a more complex generative model which better captures the statistical structure of natural stimuli[30,32,63]. Here we assumed that the generative models represented by the network match the model that generate the sensory stimuli. This is unlikely to be the case in practice. Such mismatch between the true and internal model of the world can lead to biases and increased noise which are likely to manifest in specific ways in neural circuits that perform inference via sampling[64]. Furthermore, we only considered sampling driven by spiking variability with a Fano factor of 1, while cortical responses often have Fano factors that differ from 1[65,66]. In the latter case, our theory can still work by changing the feedforward connection weight to compensate for the change in Fano factor, as suggested in a recent study[67].

To keep our exposition transparent, we only presented models with minimal complexity. Our proposed network mechanism of sampling-based inference can be generalized to more complex generative models, since the assumption of Gaussianity (Eqs. (21) and (22)) and the analytical expression in Eq. (24) are not essential, and several relaxed frameworks may be explored. First, similar networks can generate samples from other multi-dimensional distributions where the conditional distribution of each latent variable belongs to the linear exponential family[38,39]. This could be done by changing the tuning functions of neurons to another appropriate profile, as the logarithm of tuning determines the type of sampling distribution (Eq.

(1)). When sampling from non-Gaussian distributions, the stimulus samples can be linearly read out with the weight determined by the tuning profile (i.e., $\mathbf{h}(s)$ in Eq. (1)[39],). Second, the tuning of recurrent inputs does not need to be the same as that of feedforward inputs. Instead, the logarithm of recurrent input tuning can have a form of the conjugate prior with the likelihood conveyed by feedforward inputs. Third, the network model could also be used to infer the latent variables with a non-uniform marginal prior, if, for example, the preferred stimuli of neurons in the population are not distributed uniformly in the stimulus subspace[68]. And the proposed network model has the potential to produce samples from the posterior distribution of latent dynamic stimuli which can be described by a hidden Markov model. Lastly, we considered only non-structured inhibition for simplicity. Structured inhibitory connections could modulate the position of excitatory responses in the stimulus subspace, i.e., the mean of the conditional distribution. Such interplay between E and I neurons with structured inhibition has the potential to implement Hamiltonian sampling, where the I neurons represent the sample of auxiliary variables[38,50].

In conclusion, we have shown that a recurrent circuit of neurons with Poisson spiking statistics can implement sampling from a family of multivariate posterior distributions, with internal spiking variability driving the generation of stimulus samples, and the recurrent connections representing the stimulus prior. The proposed neural code may help us understand the structure of neuronal activity, and provide a building block for more complicated population computations.

## Methods

### A linear network of excitatory neurons

We study how a generic recurrent network model consisting solely of $N_E$ excitatory (E) neurons with Poisson spiking statistics (no inhibitory neurons) can implement sampling-based Bayesian inference to approximate the stimulus posterior. We describe neuronal activity using a time-discretized Hawkes process (a type of multivariate, inhomogeneous Poisson process[69]). The instantaneous firing rates of the neurons in the network at time $t$, $\boldsymbol{\lambda}_t$, obey the following recurrent equations:

$$\boldsymbol{\lambda}_t \Delta t = \mathbf{u}^f + \mathbf{u}_t^r = \mathbf{u}^f + (w_E \mathbf{r}_{t-\Delta t} + \sigma_r \boldsymbol{\xi}_t), \qquad (10)$$

$$\mathbf{r}_t \sim \prod_{j=1}^{N_E} \text{Poisson}\left(\boldsymbol{\lambda}_{tj} \Delta t\right), \qquad (11)$$

where $\mathbf{u}^f$ is the feedforward Poisson spiking input (described below; Eq. (18)), $\mathbf{u}_t^r$ is the continuous valued recurrent input at time $t$, and $\boldsymbol{\xi}_t$ is a $N_E$ dimensional independent Gaussian white noise. Hence, over each time interval $[t - \Delta t, t]$ the activity of the neurons in the network is modeled by a vector of independently generated Poisson spike counts, $\mathbf{r}_t$, with means determined by the rates $\boldsymbol{\lambda}_t$. The parameters $w_E$ and $\sigma_r$ determine the excitatory recurrent weight and recurrent variability, respectively. The instantaneous firing rate $\boldsymbol{\lambda}_t$ can be negative due to the recurrent input and noise (Eq. (36)). We interpret a negative firing rate, $\boldsymbol{\lambda}_t$, as a zero probability of generating a spike.

### Poisson spike generation samples stimulus

Independent Poisson spike generation in the network whose activity is described by Eq. (11) can drive sampling across time or across trials from a conditional stimulus distribution determined by the instantaneous firing rate $\boldsymbol{\lambda}_t$. Below, we compute the distribution of stimulus samples given $\boldsymbol{\lambda}_t$. We assume that the instantaneous firing rate, $\boldsymbol{\lambda}_t$, has a smooth bell-shaped profile and can be parameterized as,

$$\boldsymbol{\lambda}_{tj} = R \exp[-(\bar{s}_t - \theta_j)^2/2a^2] = R \exp[\mathbf{h}_j(\bar{s}_t)], \qquad (12)$$

where $\bar{s}_t$ characterizes the position of the population firing rate on the stimulus subspace (Fig. 1b, x-axis), while $R$ and $a$ denote the height and width of the population firing rate, respectively. Further, $\theta_j$ is the preferred stimulus value of neuron $j$, and the preferred stimuli of all neurons, $\{\theta_j\}_{j=1}^{N_E}$, are uniformly distributed over the range of stimulus $s$ (Fig. 1b).

To simplify the analysis, we first assume that the instantaneous firing rate is fixed over time. When generating Poisson spikes $\mathbf{r}_t$ from $\boldsymbol{\lambda}_t$, the probability of observing a stimulus sample $\tilde{s}_t$ (embedded in $\mathbf{r}_t$) can be derived as (see details in Supplementary Information),

$$
\begin{aligned}
p(\mathbf{r}_t|\boldsymbol{\lambda}_t) &= \prod_{j=1}^{N_E} \text{Poisson}\left(\mathbf{r}_{tj}|\lambda_{tj}\Delta t\right), \\
&\propto \exp[\mathbf{h}(\bar{s}_t)^\top \mathbf{r}] \cdot \left[n_{\boldsymbol{\lambda}}^{n_{\mathbf{r}}} \exp(-n_{\boldsymbol{\lambda}})\right], \\
&\propto \mathcal{N}\left(\tilde{s}_t|\bar{s}_t, a^2 n_{\mathbf{r}}^{-1}\right) \text{Poisson}(n_{\mathbf{r}}|n_{\boldsymbol{\lambda}}),
\end{aligned}
\tag{13}
$$

where $n_{\mathbf{r}} = \sum_j \mathbf{r}_{tj}$ is the number of emitted spikes across the whole neural population, and $n_{\boldsymbol{\lambda}} = \sum_j \langle \lambda_j \rangle \Delta t$ is the sum of population firing rate. Here $\mathcal{N}(s|\mu,\sigma^2)$ denotes a Gaussian distribution with mean $\mu$ and variance $\sigma^2$, and $\mathbf{h}(\bar{s}_t)$ is a vector with the $j$th element as $\mathbf{h}_j(\bar{s}_t)$ shown in Eq. (12). The logarithm of the firing rate profile, $\mathbf{h}(\bar{s}_t)$, determines how the stimulus sample $\tilde{s}_t$ and its mean, $\bar{s}_t$, can be read out respectively from $\mathbf{r}_t$ and $\boldsymbol{\lambda}_t$,

$$
\tilde{s}_t = \sum_j \mathbf{r}_{tj}\theta_j / \sum_j \mathbf{r}_{tj}, \quad \bar{s}_t = \sum_j \lambda_{tj}\theta_j / \sum_j \lambda_{tj},
\tag{14}
$$

where $\tilde{s}_t$ and $\bar{s}_t$ characterizes the position of $\mathbf{r}_t$ and $\boldsymbol{\lambda}_t$ on the stimulus subspace.

The sampling variability of $\tilde{s}_t$ in a single time step depends on the number of emitted spikes, $n_{\mathbf{r}}$. When the fixed rates, $\boldsymbol{\lambda}_t$, repeatedly generate spikes over time, the sampling distribution of $\tilde{s}_t$ can be calculated by marginalizing the likelihood (Eq. (13), last line) over different values of $n_{\mathbf{r}}$ since $n_{\mathbf{r}}$ varies across time (detailed calculation by using Laplacian approximation can be seen in Supplementary Information),

$$
\begin{aligned}
p(\tilde{s}_t|\boldsymbol{\lambda}_t) &= \sum_{n_{\mathbf{r}}} \mathcal{N}\left(\tilde{s}_t|\bar{s}_t, a^2 n_{\mathbf{r}}^{-1}\right) \text{Poisson}(n_{\mathbf{r}}|n_{\boldsymbol{\lambda}}), \\
&\approx \mathcal{N}\left(\tilde{s}_t|\bar{s}_t, a^2 n_{\boldsymbol{\lambda}}^{-1}\right).
\end{aligned}
\tag{15}
$$

Each stimulus sample, $\tilde{s}_t$, is thus drawn from a conditional distribution determined by the instantaneous firing rate, $p(\tilde{s}|\boldsymbol{\lambda}_t)$, and can be written as

$$
\tilde{s}_t \sim p(\tilde{s}|\boldsymbol{\lambda}_t) = \mathcal{N}\left(\tilde{s}|\bar{s}_t, a^2 n_{\boldsymbol{\lambda}}^{-1}\right) \propto \exp[\mathbf{h}(\tilde{s})^\top \boldsymbol{\lambda}_t].
\tag{16}
$$

The last proportionality in the above equation is satisfied by a Gaussian profile in the firing rate (more general derivation can be found in Supplementary Information). Introducing $\Lambda = a^{-2} n_{\boldsymbol{\lambda}}$ gives Eq. (1) shown in the main text.

Eq. (16) suggests that the type of sampling distribution (or the conditional distribution) that is obtained from spike generation variability is determined by the profile of the instantaneous firing rate, i.e., $\mathbf{h}(\bar{s}_t)$ (Eq. (12)). Although the sampling distribution belongs to the linear exponential family of distributions which is similar to the probabilistic population code (PPC)[39], there are different ways in representing these distributions. In PPCs, the likelihood over $\bar{s}_t$ is parametrically represented by a single realization of independent neuronal response $\mathbf{r}$ (Eq. (13)), while in our work the distribution is approximated by a sequence of samples, $\tilde{s}_t$, effectively generated by conditionally independent Poisson spike discharges.

The above analysis can be extended to the case where the instantaneous firing rate, $\boldsymbol{\lambda}_t$, in a time step deviates from a smooth Gaussian profile (Eq. (12)), which is the case in the actual network simulations. In general, $\boldsymbol{\lambda}_t$ can be expressed as,

$$
\lambda_{tj} = R_t \exp[\mathbf{h}_j(\bar{s}_t)] + \delta_\perp \lambda_{tj},
\tag{17}
$$

where $\delta_\perp \lambda_t$ denotes the deviation from a smooth Gaussian profile. Note that the sampling distribution only depends on the position, $\bar{s}_t$, and the sum of instantaneous firing rate, $n_{\boldsymbol{\lambda}}$ (Eq. (16)), which corresponds to two perpendicular directions in the $N_E$ dimensional space of $\boldsymbol{\lambda}_t$. For any instantaneous firing rate vector, $\boldsymbol{\lambda}_t$, we can always find $\bar{s}_t$ and $R_t$ that make the deviation $\delta_\perp \lambda_t$ perpendicular to the two directions, i.e., $\sum_j \delta_\perp \lambda_{tj}\theta_j = 0$, and $\sum_j \delta_\perp \lambda_{tj} = 0$. This observation implies that deviations from Gaussian firing rate profiles do not affect our theory.

## Feedforward spiking input conveys the likelihood of stimulus

We model the feedforward inputs to the E neurons in the network, $\mathbf{u}^f$, as independent Poisson spikes, with Gaussian tuning over stimulus $s$,

$$
\begin{aligned}
p(\mathbf{u}^f|s) &= \prod_{j=1}^{N_E} \text{Poisson}\left[\mathbf{u}_j^f|\langle \mathbf{u}_j^f(s) \rangle\right], \\
\langle \mathbf{u}_j^f(s) \rangle &= U^f \exp[\mathbf{h}_j(s)] = U^f \exp[-(\theta_j - s)^2 / 2a^2].
\end{aligned}
\tag{18}
$$

Here $\mathbf{u}_j^f$ denotes the feedforward input received by the $j$th E neuron, and $\langle \mathbf{u}_j^f(s) \rangle$ is the tuning of the feedforward input. This mathematical description of feedforward input is the same as the one used in the definition of typical PPCs[15,39,40]. Since the preferred stimulus values, $\{\theta_j\}_{j=1}^{N_E}$, of all feedforward inputs are uniformly distributed in stimulus space then the likelihood of $s$ given a single observation of the input, $\mathbf{u}^f$, satisfies[39,40],

$$
\begin{aligned}
p(\mathbf{u}^f|s) &\propto \exp\left[\mathbf{h}(s)^\top \mathbf{u}^f\right], \\
&\propto \mathcal{N}\left(s|\mu_f, \Lambda_f^{-1}\right).
\end{aligned}
\tag{19}
$$

The logarithm of tuning, $\mathbf{h}(s)$, determines the type of likelihood[15]. Specifically, the Gaussian tuning leads to a Gaussian likelihood (Eq. (19)), whose mean, $\mu_f$, and precision, $\Lambda_f$, are both linear functions of the inputs,

$$
\mu_f = n_f^{-1} \sum_j \mathbf{u}_j^f \theta_j, \quad \Lambda_f = a^{-2} n_f = a^{-2} \sum_j \mathbf{u}_j^f.
\tag{20}
$$

The mean, $\mu_f$, represents the position of $\mathbf{u}^f$ in stimulus subspace, and the precision, $\Lambda_f$, is proportional to the sum of total feedforward spike counts, $n_f$.

## A recurrent network samples hierarchical latent variables

**A hierarchical generative model.** We consider a hierarchical generative model for which inference can be implemented in a recurrent circuit of Poisson neurons. We extend the simple generative model of feedforward input (Eq. (19)) by considering the stimulus $s$ to depend on a one-dimensional stimulus parameter variable, $z$. For simplicity, we assume that $z$ follows a uniform distribution (Fig. 3b, marginal plots)

$$
p(z) = \mathcal{U}(-180°, 180°),
\tag{21}
$$

where $\mathcal{U}(a,b)$ denotes a uniform distribution over $[a, b]$. The assumption of a uniform prior, $p(z)$, simplifies our model significantly, as it implies the spatial homogeneity of the network model as given by Eqs. ((18), (19)). However, this assumption is not essential for our main results. Due to the differences between the stimulus and its underlying

parameter of the sensory scene, the stimulus, $s$, is not identical to the parameter $z$, but we assume that the two are correlated, so that

$$p(s|z,\Lambda_s) = \mathcal{N}\left(s|z,\Lambda_s^{-1}\right). \tag{22}$$

In sum, the whole generative model is determined by,

$$\begin{aligned} p(\mathbf{u}^f,s,z) &= p(\mathbf{u}^f|s)p(s|z)p(z), \\ &\propto \mathcal{N}\left(s|\mu_f,\Lambda_f^{-1}\right)\mathcal{N}\left(s|z,\Lambda_s^{-1}\right), \end{aligned} \tag{23}$$

where $p(\mathbf{u}^f|s)$ is the same as in Eq. (19).

**Approximate Bayesian inference via Gibbs sampling.** The joint posterior of $s$ and $z$ can be analytically derived given the generative model (Eq. (23)),

$$\begin{aligned} p(s,z|\mathbf{u}^f) &= \mathcal{N}\left[(s,z)^\top|\boldsymbol{\mu}_p,\mathbf{K}_p^{-1}\right], \\ \boldsymbol{\mu}_p &= (\mu_f,\mu_f)^\top, \quad \mathbf{K}_p = \begin{pmatrix} \Lambda_f+\Lambda_s & -\Lambda_s \\ -\Lambda_s & \Lambda_s \end{pmatrix}. \end{aligned} \tag{24}$$

We will use this expression to verify that the samples produced by our algorithm converge to the output of the algorithm.

We use the stochastic response of our recurrent network (Eqs. (10), (11)), as a basis for Gibbs sampling[31,38,42] (a type of Monte Carlo method) to approximate the joint posterior $p(s,z)$. To describe the iterative Gibbs algorithm, we assume that a stimulus parameter sample, $\tilde{z}_t$, is provided at time $t$, which is then combined with the feedforward input to update the conditional distribution of stimulus $s$ (step 1 in Fig. 3c),

$$\begin{aligned} p(\tilde{s}|\tilde{z}_t,\mathbf{u}^f) &\propto p(\mathbf{u}^f|\tilde{s})p(\tilde{s}|\tilde{z}_t) \propto \mathcal{N}\left(s|\bar{s}_t,\Lambda^{-1}\right), \\ \bar{s}_t &= \frac{\Lambda_f\mu_f+\Lambda_s\tilde{z}_t}{\Lambda_f+\Lambda_s}, \quad \Lambda = \Lambda_f+\Lambda_s. \end{aligned} \tag{25}$$

The next step in the algorithm is to draw a sample, $\tilde{s}_t$, from the conditional distribution $p(\tilde{s}|\tilde{z}_t,\mathbf{u}^f)$ (step 2 in Fig. 3c),

$$\tilde{s}_t \sim p(\tilde{s}|\tilde{z}_t,\mathbf{u}^f) = \mathcal{N}\left(\tilde{s}|\bar{s}_t,\Lambda^{-1}\right).$$

Next, the conditional distribution of stimulus parameter, $z$, is updated given this new sample, $\tilde{s}_t$, and a new sample, $\tilde{z}_{t+\Delta t}$, is drawn (step 3 in Fig. 3c),

$$\tilde{z}_{t+\Delta t} \sim p(\tilde{z}|\tilde{s}_t) = \mathcal{N}(\tilde{z}|\bar{s}_t,\Lambda_s^{-1}). \tag{26}$$

These three steps in the Gibbs sampling algorithm (Eqs. (25), (26)) are performed iteratively until sufficiently many samples, $\tilde{s}_t$ and $\tilde{z}_t$, are generated to approximate the true posterior distribution with sufficient accuracy (Fig. 3d; compare the red dots with the blue contour map).

**Implementing the Gibbs sampling in a recurrent circuit model.** Gibbs sampling of the stimulus (Eq. (4b)) can be implemented via independent Poisson spike generation, as long as the conditional distribution encoded in $\boldsymbol{\lambda}_t$ (Eq. (16)) is the same as the conditional distribution in the Gibbs sampling algorithm (Eq. (4a)), i.e., $\ln p(\tilde{s}|\boldsymbol{\lambda}_t) = \mathbf{h}(\tilde{s})^\top\boldsymbol{\lambda}_t = \ln p(\tilde{s}|\tilde{z}_t,\mathbf{u}^f)$. This condition can be realized in the recurrent circuit by relating the expressions describing the neural dynamics (Eq. (10)) and those describing the Gibbs sampling

distribution (Eq. (4a)) to yield,

$$\begin{aligned} \ln p(\tilde{s}|\tilde{z}_t,\mathbf{u}^f) &= \mathbf{h}(\tilde{s})^\top\boldsymbol{\lambda}_t, \\ &= \mathbf{h}(\tilde{s})^\top\mathbf{u}^f + \mathbf{h}(\tilde{s})^\top\mathbf{u}_t^r, \\ &= \ln p(\mathbf{u}^f|\tilde{s}) + \ln p(\tilde{s}|\tilde{z}_t). \end{aligned} \tag{27}$$

The generative model for the feedforward input $\mathbf{u}^f$ (Eq. (19)) suggests that $\ln p(\mathbf{u}^f|\tilde{s}) = \mathbf{h}(\tilde{s})^\top\mathbf{u}^f$. Hence to satisfy Eq. (27) we require

$$\ln p(\tilde{s}|\tilde{z}_t) = \mathbf{h}(\tilde{s})^\top\mathbf{u}_t^r, \tag{28}$$

which implies that the recurrent input, $\mathbf{u}_t^r$, should approximately have a Gaussian profile,

$$\begin{aligned} \mathbf{u}_{tj}^r(\tilde{z}_t) &= U^r \exp[-(\theta_j-\tilde{z}_t)^2/2a^2] + \delta_\perp\mathbf{u}_{tj}^r, \\ \tilde{z}_t &= \sum_j \mathbf{u}_{tj}^r\theta_j / \sum_j \mathbf{u}_{tj}^r, \quad \Lambda_s = a^{-2}\sum_j \mathbf{u}_{tj}^r, \end{aligned} \tag{29}$$

whose position on the stimulus subspace is $\tilde{z}_t$, and the sum of input (height) is determined by $\Lambda_s$, the precision of conditional distribution $p(s|\tilde{z}_t)$. In a similar fashion to Eq. (17), $\delta_\perp\mathbf{u}_t^r$ denotes the deviation from a smooth Gaussian and is perpendicular to the direction of $\tilde{z}_t$ and $\Lambda_s$.

The optimal recurrent weight can be derived by combining Eq. (29) and Eq. (17). We notice the recurrent input, $\mathbf{u}^r$, and neuronal responses, $\mathbf{r}_t$, have the same tuning width, $a$, in a network with only E neurons. This can only be achieved if E neurons are only self-connected (Eq. (10)), as lateral connection broaden their tuning. The optimal recurrent weight generating recurrent input with appropriate strength is then,

$$w_E^* = \frac{\langle\mathbf{u}_j^r\rangle}{\langle\mathbf{r}_j\rangle} = \frac{\sum_j\langle\mathbf{u}_j^r\rangle}{\sum_j\langle\mathbf{r}_j\rangle} = \frac{\sum_j\langle\mathbf{u}_j^r\rangle}{\sum_j\left(\langle\mathbf{u}_j^f\rangle+\langle\mathbf{u}_j^r\rangle\right)} = \frac{\Lambda_s}{\Lambda_f+\Lambda_s}, \tag{30}$$

which yields Eq. (6) in the main text. Note that the self-connection is a result of the simplifying assumption that the network consists solely of E neurons (Eq. (10)), which can be relaxed in a full network consisting both E and I neurons as we show below.

The sampling of the stimulus parameter (Eq. (4c)) can be implemented through variability in the recurrent input. To do this, we include diffusive term in the recurrent interactions, $\mathbf{u}_t^r$, and we equate the variance of the fluctuations with the mean to mimic a Poisson distribution:

$$\mathbf{u}_t^r = \bar{\mathbf{u}}_t^r + \sqrt{[\bar{\mathbf{u}}_t^r]_+}\boldsymbol{\xi}_t, \quad \bar{\mathbf{u}}_t^r = w_E^*\mathbf{r}_{t-\Delta t}, \tag{31}$$

where $[\cdot]_+$ denotes negative rectification. Here $\boldsymbol{\xi}_t$ is a $N_E$ dimensional Gaussian white noise with $\langle\boldsymbol{\xi}_t(i)\boldsymbol{\xi}_{t'}(j)\rangle = \delta_{ij}\delta(t-t')$, $\delta_{ij}$ and $\delta(t-t')$ are Kronecker and Dirac delta functions respectively, $\bar{\mathbf{u}}_t^r$ represents the conditional distribution $p(\tilde{z}|\tilde{s}_{t-\Delta t})$, and $\mathbf{u}_t^r$ represent a stimulus parameter sample $\tilde{z}_t$ (Eq. (29)). The multiplicative variability on recurrent interaction may come from synaptic noise[37,70].

## Coupled circuits sample a multi-dimensional posterior

We consider a generative model which has multiple latent stimuli, $\mathbf{s} = (s_1,s_2,\cdots,s_m)$, which are organized in parallel (Fig. 6a). Without loss of generality, we consider the simplest case where $m=2$, and the same mechanism can be straightforwardly extended to any $m>2$. We assume the joint prior of $\mathbf{s}$ is a multivariate normal distribution,

$$\begin{aligned} p(\mathbf{s}) &= \mathcal{N}(\mathbf{s}|\boldsymbol{\mu}_s,\boldsymbol{\Lambda}_s^{-1}) \propto \exp[-\Lambda_s(s_1-s_2)^2/2], \\ &\text{with} \quad \boldsymbol{\Lambda}_s = \Lambda_s\begin{pmatrix} 1 & -1 \\ -1 & 1 \end{pmatrix}, \end{aligned} \tag{32}$$

and each stimulus $s_m$ is uniformly distributed in $(-180°, 180°]$ with periodic boundary imposed. The definition of Gaussian distribution in a circular space works well as long as the variance of the distribution is much smaller than the range of stimulus space. Here $\Lambda_s$ is the precision matrix, while the scalar variable $\Lambda_s$ ($\Lambda_s \geq 0$) characterizes the correlation between $s_1$ and $s_2$. Note that the covariance matrix $\Lambda_s^{-1}$ is not defined, and the prior (Eq. (32)) is improper. The mean, $\boldsymbol{\mu}_s$, is a free parameter, because it doesn't appear in the detailed expression of the prior (Eq. (32)), which is a consequence from the zero determinant of the precision matrix, i.e., $|\Lambda_s| = 0$. A further consequence is that the prior is not centered at $\boldsymbol{\mu}_s$, but instead has a band structure along the diagonal, and the marginal prior of each stimulus feature $p(s_m)$ ($m = 1, 2$) is uniform (Fig. 6b). The uniform marginal prior simplifies our theoretical derivation as it implies the spatial homogeneity of the network model but doesn't impact the proposed neural coding mechanism.

Each stimulus $s_m$ ($m = 1, 2$) individually generates feedforward spiking input $\mathbf{u}_m^f$, whose likelihood $p(\mathbf{u}_m^f | s_m)$ is exactly the same as Eq. (2). Combined together, the generative model is

$$p(\mathbf{u}^f | \mathbf{s})p(\mathbf{s}) = \left[ \prod_{m=1}^2 p(\mathbf{u}_m^f | s_m) \right] p(s_1, s_2),$$
$$\propto \left[ \prod_{m=1}^2 \mathcal{N}(s_m | \mu_{fm}, \Lambda_{fm}^{-1}) \right] \mathcal{N}(\mathbf{s} | \boldsymbol{\mu}_s, \Lambda_s^{-1}), \quad (33)$$
$$\propto \mathcal{N}(\mathbf{s} | \boldsymbol{\mu}_f, \Lambda_f^{-1}) \mathcal{N}(\mathbf{s} | \boldsymbol{\mu}_s, \Lambda_s^{-1}),$$

where $\boldsymbol{\mu}_f = (\mu_{f1}, \mu_{f2})^\top$, and the likelihood precision matrix $\Lambda_f = \text{diag}(\Lambda_{f1}, \Lambda_{f2})$ is a diagonal matrix.

**Gibbs sampling of the multi-dimensional posterior in a coupled neural circuit.** Given the generative model (Eq. (33)), the joint posterior of $s_1$ and $s_2$ is a bivariate normal distribution, i.e., $p(\mathbf{s} | \mathbf{u}^f) = \mathcal{N}\left( \mathbf{s} | \boldsymbol{\mu}_p, \mathbf{K}_p^{-1} \right)$, whose precision matrix $\mathbf{K}_p$ and the mean $\boldsymbol{\mu}_p$ are,

$$\mathbf{K}_p = \Lambda_f + \Lambda_s, \quad \boldsymbol{\mu}_p = \mathbf{K}_p^{-1} \Lambda_f \boldsymbol{\mu}_f. \quad (34)$$

The precision matrix of the posterior is the sum of the precision of the likelihood and the prior, implying increased reliability of the distribution after combining with the prior. Meanwhile, the posterior mean is the weighted average of the means of the two likelihoods, with the weight proportional to the precision of each likelihood. We use this expression for the posterior to evaluate the performance of the proposed sampling-based algorithm.

Using Gibbs sampling to approximate the posterior (Eq. (34)) involves the following steps:

$$\text{Compute}: p(\tilde{s}_1 | \mathbf{u}_1^f, \tilde{s}_{2,t-\Delta t}) \propto p(\mathbf{u}_1^f | \tilde{s}_1) p(\tilde{s}_{2,t-\Delta t} | \tilde{s}_1), \quad (35a)$$

$$\text{Sample}: \tilde{s}_{1t} \sim p(\tilde{s}_1 | \mathbf{u}_1^f, \tilde{s}_{2,t-\Delta t}). \quad (35b)$$

We note that we only describe the sampling from the posterior distribution of $s_1$; as samples from the posterior of $s_2$ can be obtained similarly after exchanging indices. This sampling can be implemented in a neural circuit model consisting of several coupled networks, in which each network generates samples from the posterior distribution of the corresponding stimulus. Therefore, the number of networks in the coupled circuit equals the dimension of the latent stimuli. The dynamics of the coupled neural circuit is defined by:

$$\boldsymbol{\lambda}_{1t} = \mathbf{u}_1^f + \mathbf{u}_{12,t}^r = \mathbf{u}_1^f + w_{12} \mathbf{r}_{2,t-\Delta t}, \quad (36)$$

$$\mathbf{r}_{1t} \sim \prod_{j=1}^{N_E} \text{Poisson}(\boldsymbol{\lambda}_{1t,j}), \quad (37)$$

We again note the dynamics of network 2 can be similarly obtained by changing indices. To implement Gibbs sampling (Eqs. (35a), (35b)) in the coupled circuit (Eqs. (36), (37)), spike generation in network 1 (Eq. (37)) can be used to produce stimulus samples, $\tilde{s}_{1t}$, when the conditional distribution determined by $\boldsymbol{\lambda}_{1t}$ matches the conditional distribution required in the definition of Gibbs sampling (Eq. (35a)), i.e., $\ln p(\tilde{s}_1 | \mathbf{u}_1^f, \tilde{s}_{2,t-\Delta t}) = \ln p(\tilde{s}_{1t} | \boldsymbol{\lambda}_{1t}) = \mathbf{h}(\tilde{s}_1)^\top \boldsymbol{\lambda}_{1t}$. Taking the logarithm of Eq. (35a) yields,

$$\ln p(\tilde{s}_1 | \mathbf{u}_1^f, \tilde{s}_{2,t-\Delta t}) = \ln p(\mathbf{u}_1^f | \tilde{s}_1) + \ln p(\tilde{s}_{2,t-\Delta t} | \tilde{s}_1). \quad (38)$$

Comparing this expression with Eq. (36), we see that the feedforward input, $\mathbf{u}_1^f$, matches the conditional distribution $p(\mathbf{u}_1^f | \tilde{s}_1)$ (Eq. (33)). We therefore require the recurrent input from network 2 to network 1 to encode the conditional distribution $p(\tilde{s}_{2,t-\Delta t} | \tilde{s}_1)$, i.e., $\ln p(\tilde{s}_{2,t-\Delta t} | \tilde{s}_1) = \mathbf{h}(\tilde{s}_1)^\top \mathbf{u}_{12,t}^r$. This implies that $\mathbf{u}_{12,t}^r$ should approximately have a Gaussian profile,

$$\mathbf{u}_{12,tj}^r = U_{12}^r \exp[-(\theta_j - \tilde{s}_{t-\Delta t})^2 / 2a^2] + \delta_\perp \mathbf{u}_{12,tj}^r,$$
$$\tilde{s}_{2,t-\Delta t} = \sum_j \mathbf{u}_{12,tj}^r \theta_j / \sum_j \mathbf{u}_{12,tj}^r, \quad \Lambda_s = a^{-2} \sum_j \mathbf{u}_{12,tj}^r, \quad (39)$$

where $\delta_\perp \mathbf{u}_{12,tj}^r$ quantifies the deviation from a perfect Gaussian profile, and does not affect the decoded value $\tilde{s}_{2,t-\Delta t}$ and $\Lambda_s$.

The recurrent input, $\mathbf{u}_{12}^r$, (Eq. (39)) has the same width $a$ as the neuronal response, $\mathbf{r}_1$. In circuit containing only E neurons, if the two networks have the same number of neurons, then across networks only neurons having the same preferred stimulus should be connected. The optimal recurrent weight between two networks is then

$$w_{mn} = \frac{\langle \mathbf{u}_{mn,j}^r \rangle}{\langle \mathbf{r}_{nj} \rangle} = \frac{\sum_j \langle \mathbf{u}_{mn,j}^r \rangle}{\sum_j \langle \mathbf{r}_{nj} \rangle} = \frac{\Lambda_s}{\Lambda_s + \Lambda_n^f}, \ (m \neq n) \quad (40)$$

Since each network individually generate a stimulus sample, the sample of stimulus $m$ can be locally read out from network $m$'s responses even if the activities of two networks are correlated (Fig. 6a), which greatly simplifies readout. Furthermore, due to the population firing rate of each network has Gaussian profile, the stimulus sample $\tilde{s}_{mt}$ can be linearly read out from $\mathbf{r}_{mt}$ as

$$\tilde{s}_{mt} = \sum_j \theta_j \mathbf{r}_{mt,j} / \sum_j \mathbf{r}_{mt,j}. \quad (41)$$

We note that the circuit implementation of Gibbs sampling from a multi-dimensional posterior (Eq. (8a)) does not require the recurrent connections between E neurons within a network. This is due to the assumption that the marginal priors of each stimulus feature, $p(s_m)$, are uniform. For a non-uniform marginal prior $p(s_m)$, recurrent connections between E neurons within a network would be required for generating samples from a distribution that matches the true posterior.

## Inference from an information-theoretic point of view

The goal of the sampling algorithm is to approximate the posterior distribution of a latent variables, $\Theta$, given a feedforward input, $\mathbf{u}^f$. Specifically, the latent variables $\Theta = \{s, z\}$ in the hierarchical generative model (Eq. (23)), or $\Theta = \mathbf{s} = \{s_1, s_2\}$ in the generative model with breadth (Eq. (33)). When the sampling algorithm uses an internal model which does not match the structure of the generative model, the sampling

distribution $q(\Theta|\mathbf{u}^f)$ will differ from the true posterior, $p(\Theta|\mathbf{u}^f)$ (Eq. (24)). In this case the mutual information between the sampling distribution of the latent variables, $\Theta$, and $\mathbf{u}^f$ will be smaller than in the case when samples come from the true posterior, $p(\Theta|\mathbf{u}^f)$,

$$
\begin{aligned}
I(\Theta,\mathbf{u}^f) &= -\mathbb{E}_{p(\Theta)}[\log p(\Theta)] + \mathbb{E}_{p(\Theta,\mathbf{u}^f)}[\log p(\Theta|\mathbf{u}^f)] \\
&\geq -\mathbb{E}_{p(\Theta)}[\log p(\Theta)] + \mathbb{E}_{p(\Theta,\mathbf{u}^f)}[\log q(\Theta|\mathbf{u}^f)] \equiv I_q(\Theta;\mathbf{u}^f),
\end{aligned}
\tag{42}
$$

It is straightforward to show that the difference between $I(\Theta,\mathbf{u}^f)$ and $I_q(\Theta,\mathbf{u}^f)$ is the Kullback–Leibler (KL) divergence between $p$ and $q$, i.e., $D_{KL}[p||q] = I(\Theta,\mathbf{u}^f) - I_q(\Theta,\mathbf{u}^f) = \mathbb{E}_p(\ln p - \ln q) \geq 0$. Equality in Eq. (42) holds only if the distribution $q$ matches the true posterior $p$.

The mutual information $I_q(\Theta;\mathbf{u}^f)$ can be computed analytically when the approximating distribution $q(\Theta|\mathbf{u}^f) = \mathcal{N}(\Theta|\boldsymbol{\mu}_q,\mathbf{K}_q^{-1})$ is a bivariate normal (substituting Eqs. (23) and (24) into Eq. (42)),

$$
I_q(\Theta;\mathbf{u}^f) = \log L + \frac{1}{2}\left[1 + \log\frac{|\mathbf{K}_q|}{2\pi\Lambda_s} - \mathrm{tr}(\mathbf{K}_q\mathbf{K}_p^{-1}) - (\boldsymbol{\mu}_p - \boldsymbol{\mu}_q)^\top \mathbf{K}_q(\boldsymbol{\mu}_p - \boldsymbol{\mu}_q)\right]. \tag{43}
$$

Here $L = 360°$ is the length of the stimulus feature subspace, while $\boldsymbol{\mu}_p$ and $\mathbf{K}_p$ are the mean and the precision matrix of the posterior distribution (Eqs. (24) or (34)). When $q$ matches the posterior distribution, $p$, we have, $I(\Theta;\mathbf{u}^f) = \log L - \frac{1}{2}[1 + \log(2\pi\Lambda_s) - \log|\mathbf{K}_p|]$.

## The neuronal response distribution conditioned on external stimulus

We compute the distribution of neuronal responses $\mathbf{r}$ over time/trial in response to an external stimulus $s$, i.e., $p(\mathbf{r}|s)$, in order to find a neural signature of network sampling and compare it with experimental data. For a fixed external stimulus $s$, the neuronal response $\mathbf{r}$ fluctuates due to both sensory transmission noise described by $p(\mathbf{u}^f|s)$ (Eq. (18)), as well as the internally generated variability described by $p(\mathbf{r}|\mathbf{u}^f)$ (Fig. 4a). Therefore, the distribution of $\mathbf{r}$ in response to an external stimulus $s$ has the form

$$
p(\mathbf{r}|s) = \int p(\mathbf{r}|\mathbf{u}^f)p(\mathbf{u}^f|s)d\mathbf{u}^f.
$$

For simplicity, we only compute the covariability of $p(\mathbf{r}|\mathbf{u}^f)$ along the stimulus subspace (Fig. 1b, x-axis), because the covariability along other directions is not related with stimulus sampling. By approximating the Poissonian spiking variability $p(\mathbf{r}|\boldsymbol{\lambda})$ with a multivariate normal distribution (Eq. (11)), and considering the limit of weak fluctuations in $\boldsymbol{\lambda}$ along the stimulus subspace over time, $p(\mathbf{r}|\mathbf{u}^f)$ can be computed approximately as (see math details in Supplementary Information),

$$
\begin{aligned}
p(\mathbf{r}|\mathbf{u}^f) &= \int p(\mathbf{r}|\boldsymbol{\lambda})p(\boldsymbol{\lambda}|\mathbf{u}^f)d\boldsymbol{\lambda}, \\
&\approx \mathcal{N}\left[\mathbf{r}|\mathbf{f}(s),\mathrm{diag}(\mathbf{f}(s)) + V(\bar{s}|\mu_f)\mathbf{f}'_s\mathbf{f}'^\top_s\right], \quad \text{where } s = \mu_f.
\end{aligned}
\tag{44}
$$

$\mathbf{f}(s) = \langle\boldsymbol{\lambda}_t\rangle$ denotes the temporally averaged population response. The covariance structure of the neuronal response includes two terms: $\mathrm{diag}(\mathbf{f}(s))$, a diagonal matrix whose entries equal that of the vector $\mathbf{f}(s)$ denoting the (independent) Poisson spiking variability (Eq. (23)), and $V(\bar{s}|\mu_f)\mathbf{f}'_s\mathbf{f}'^\top_s$, a term that captures the covariability due to firing rate fluctuations along the stimulus subspace (Fig. 8a), where $\mathbf{f}'_s = d\mathbf{f}(s)/ds$ is the derivative of $\mathbf{f}(s)$ over the stimulus feature $s$. The covariance $\mathbf{f}'_s\mathbf{f}'^\top_s$ is often termed differential (noise) correlations[4,17]. With the Gaussian profile of $\mathbf{f}(s)$ (Eqs. (18) and (29)), $\mathbf{f}'_s\mathbf{f}'^\top_s$ exhibits anti-symmetric structure (Fig. 8b)[17,22,53,71,72].

In Eq. (44), $V(\bar{s}|\mu_f)$ is the variance of $\bar{s}_t$ (the mean of conditional distribution in Eq. (4a)) over time and characterizes the amplitude of internally generated differential correlations. In network implementation, $\bar{s}_t$ and $\mu_f$ are represented as the position of $\boldsymbol{\lambda}_t$ and $\mathbf{u}^f$ on the

stimulus subspace respectively (Eqs. (14) and (20)). The dynamics of Gibbs sampling (Eq. S20 in Supplementary Information) and the network structure (Eq. (6)) imply that

$$
V(\bar{s}|\mu_f) = \frac{\Lambda_s}{\Lambda_f(\Lambda_f + \Lambda_s)} = a^2 n_f^{-1} w_E^*. \tag{45}
$$

Note that $V(\bar{s}|\mu_f)$ is constrained by network connections, in that it is internally generated and shared within the network (for $w_E^* > 0$).

An expression for $p(\mathbf{r}|s)$ can be derived similarly, and includes an additional term contributing to differential correlations compared with $p(\mathbf{r}|\mathbf{u}^f)$ (Eq. (44)) due to fluctuations in the feedforward inputs,

$$
\begin{aligned}
p(\mathbf{r}|s) &\approx \mathcal{N}\left[\mathbf{r}|\mathbf{f}(s),\mathrm{diag}(\mathbf{f}(s)) + V(\bar{s}|s)\mathbf{f}'_s\mathbf{f}'^\top_s\right], \\
V(\bar{s}|s) &= V(\bar{s}|\mu_f) + V(\mu_f|s) = \frac{\Lambda_s}{\Lambda_f(\Lambda_f + \Lambda_s)} + \frac{1}{\Lambda_f} = a^2 n_f^{-1}(w_E^* + 1).
\end{aligned}
\tag{46}
$$

Here the variance, $V(\bar{s}|s)$, in the stimulus feature subspace is a mixture of internal variability, $V(\bar{s}|\mu_f)$, and sensory noise, $V(\mu_f|s)$ (Eq. (23)). The neuronal response distribution in coupled networks (Fig. 6a) can be obtained similarly (see the Supplementary Information).

## A spiking network model with excitatory and inhibitory Poisson neurons

To test the proposed inference mechanisms in a network consisting of E neurons (Eqs. (10)–(37)), we simulated a well studied recurrently coupled cortical model[21,22]. The network consisted of $N_E$ excitatory (E) and $N_I$ inhibitory (I) spiking neurons, with the activity of each neuron modeled as a Hawkes process[69]. At time $t$, we represent the response of neuron $j$ in population $a = \{E, I\}$, $\mathbf{r}_{tj}^a$, as a spike count drawn from a Poisson distribution with instantaneous firing rate, $\boldsymbol{\lambda}_{tj}^a$,

$$
\mathbf{r}_{tj}^a \sim \mathrm{Poisson}\left[\boldsymbol{\lambda}_{tj}^a\right]. \tag{47}
$$

Each neuron has a refractory period of 2ms after emitting a spike. The firing rate $\boldsymbol{\lambda}_{tj}^a$ is the sum of feedforward input $\mathbf{u}_{tj}^{af}$ and recurrent input $\mathbf{u}_{tj}^{ar}$, so that $\boldsymbol{\lambda}_{tj}^a = \mathbf{u}_{tj}^{af} + \mathbf{u}_{tj}^{ar}$. The feedforward inputs are filtered spikes from upstream neurons, $\mathbf{u}_{tj}^{af} = \sum_n \eta\left(t - t_{jn}^f\right)$, where $t_{jn}^a$ is the time of the $n$th spike received by neuron $j$ of population $a$ from the feedforward inputs. Here $\eta(t)$ is the synaptic input profile which is modeled as $\eta(t) = \exp(-t/\tau_d)/\tau_d$, $(t > 0)$. Throughout, we set the synaptic time constant $\tau_d = 2$ms. To mimic the Poisson-like variability to sample a stimulus parameter in a hierarchical generative model (Eqs. (23) and (31)), the recurrent input received by neuron $j$ in population $a$ is defined by

$$
\begin{aligned}
\mathbf{u}_{tj}^{ar} &= \bar{\mathbf{u}}_{tj}^{ar} + \sqrt{[\bar{\mathbf{u}}_{tj}^{ar}]_+}\,\xi_t, \\
\bar{\mathbf{u}}_{tj}^{ar} &= \sum_{b=\{E,I\}}\sum_{k=1}^{N_b}\frac{J_{jk}^{ab}}{\sqrt{N}}\sum_n \eta(t - t_{kn}^b),
\end{aligned}
\tag{48}
$$

where $\bar{\mathbf{u}}_{tj}^{ar}$ is the mean recurrent input at time $t$ given the neuronal activities of the presynaptic neurons. The recurrent input in the network is corrupted by noise whose variance equals the mean of the recurrent input. In a physiological network, recurrent noise may be generated by the chaotic state in network dynamics[36] or synaptic noise[37,70]. In Eq. (48), the function $[\cdot]_+$ rectifies the negative input, and $\xi_t$ is a random variable following a standard Gaussian distribution. The coefficient $J_{ij}^{ab}$ is the synaptic weight from neuron $j$ in population $b$ to neuron $i$ in population $a$. The time $t_{kn}^b$ is the time of the $n$th spike fired by neuron $k$ in population $b$. The parameter $N = N_E + N_I$ is the total number of neurons in the network. The scaling of the synaptic weights by $1/\sqrt{N}$ is standard in networks where excitation is balanced by recurrent inhibition[36]. Finally, the synaptic input profile of the

 

recurrent input, $\eta(t)$, is the same as the one we chose for the feedforward input for convenience. Note that the rectification in Eq. (48) on recurrent inputs will introduce errors resulting in deviations of the sampling distribution from the true posterior, and hence we chose the recurrent weights to be small (Fig. 5). The rectification only arises when using (continuous) recurrent inputs to sample the stimulus parameter, and doesn't impact the generality of sampling by (discrete) Poisson spiking variability.

To model the coding of a circular stimulus such as orientation, the excitatory neurons are arranged on a ring[22,71]. The preferred stimuli, $\theta_j$, of the excitatory neurons are equally spaced on the interval $(-180°, 180°]$, consistent with the range of latent features (Eq. (21)). Inhibitory neurons are not tuned to stimulus, and their role is to stabilize network responses. Note that the recurrent connections between $E$ neurons are modeled using a Gaussian function decaying with the distance between the stimuli preferred by the two cells, rather than only self-connection in the simple network with only E neurons (Eq. (30)),

$$J_{jk}^{EE} = \frac{w_{EE}L}{\sqrt{2\pi}a}\exp\left[-\frac{(\theta_j - \theta_k)^2}{2a^2}\right], \tag{49}$$

We imposed periodic boundaries on the Gaussian function to avoid boundary effect in simulations. Although in the generative model we assumed non-periodic feature variables (Eq. (3)), as long as the variance of the associated distributions are smaller than the width of the feature space, the network model with periodic boundaries on the recurrent connection (Eq. (49)) provides a good approximation of the non-periodic Gaussian posterior (Eq. (24)). The weight $w_{EE}$ denotes the average connection strength of all $E$ to $E$ connections. The parameter $a = 40°$ defines the footprint of connectivity in feature space (i.e the ring), and $L = 360°$ is the length of the ring manifold (Eq. (21)); Multiplication by $L$ in Eq. (49) sets the sum of all $E$ to $E$ connection strengths equal to $N_E w_{EE}$. Moreover, the excitatory and inhibitory neurons are all-to-all connected with each other (similar for $I$ to $I$ connections). For simplicity, we consider the $E$ to $I$, $I$ to $I$ and $I$ to $E$ connections all to be unstructured (in feature space) and assume that connections of the same type have equal weight, i.e., $J_{jk}^{EI} = w_{EI}$, $J_{jk}^{IE} = w_{EE}$ and $J_{jk}^{II} = w_{II}$. To simplify the network further, we consider the connections from the same population of neurons to have the same average weight, i.e., $w_{EE} = w_{IE} \equiv w_E$ and $w_{II} = w_{EI} \equiv w_I$. For the feedforward network model shown in Fig. 2, we only remove the E recurrent connections between E neurons, i.e., $w_{EE} = 0$, while keeping other connections, including $w_{EI}$, $w_{II}$, and $w_{IE}$, the same as the recurrent network.

The feedforward inputs applied to E neurons consist of independent Poisson spike counts as described by Eq. (18), with rate $\langle \mathbf{u}_j^{Ef}(s) \rangle = U^f e^{-(s-\theta_j)^2/(4a^2)}$. The inhibitory neurons also receive feedforward indpendent Poissonian inputs. The firing rate of the input received by every $I$ neuorn is proportional to the overall feedforward rate of input to $E$ neurons, in order to keep the excitatory and inhibitory balance of neuronal activities in the network,

$$\langle \mathbf{u}_j^{If} \rangle = \frac{w_{If}}{N_I}\sum_{j=1}^{N_E}\langle \mathbf{u}_j^{Ef}(s)\rangle. \tag{50}$$

In the simulations, we started with a network of $N_E = 180$ excitatory and $N_I = 45$ inhibitory neurons, and increased the number of neurons by a fixed factor in Fig. 1d. The ratio between the average connection from $I$ neurons and the one from $E$ neurons was kept fixed with $w_I/w_E = 5$. We set the feedforward weight of input to I neurons to $w_{fI} = 0.8$. We simulated the dynamics of the model network using the Euler method with a time step of 0.1ms. The typical parameters used in simulation can be found in Table 1 in Supplementary Information. Further details about the simulations and numerical estimates of

mutual information and linear Fisher information are also presented in Supplementary Information. The code of network simulation was written in MATLAB 2018b, and can be found at GitHub (https://github.com/wenhao-z/Sampling_PoissSpk_Neuron).

**A spiking network model of coupled neural circuits.** In the coupled neural circuits used to infer latent variables organized in parallel (Fig. 6a) the two networks are copies of each other, i.e., the two networks have the same intrinsic parameters. Each network is equivalent to the one described in the previous section, except that there is no recurrent connections between E neurons in the same network, and no variability in recurrent interactions (no noise in Eq. (48)). The absence of recurrent connections between E neurons in the same network is due to the uniform marginal prior of stimulus. Nevertheless, in the same network the E and I neurons are connected using the same connection profile as above to keep network activity stable. Between the two networks, there are only E connections which target both E and I neurons. The connections between E neurons across networks have the same pattern as that given described by Eq. (49) with the peak connection strength from network $n$ to network $m$ denoted as $w_{EE}^{mn}$. The connections from E neurons in one network to I neurons in the other is set to the same as the peak strength of E connections across networks for simplicity, i.e., $w_{IE}^{mn} = w_{EE}^{mn}$. To simplify the network model further, we set the inter-network connections to be symmetric, which means $w_{EE}^{mn} = w_{EE}^{nm}$. In the simulations $w_{EE}^{mn}$ was adjusted to determine how the sampling distribution is affected (Fig. 7a).

**Comparing the sampling distribution with posterior in coupled neural circuits.** We read out the samples from the posterior distribution of each stimulus, $\tilde{s}_{mt}$, individually from the spiking activities of E neurons, $\mathbf{r}_{mt}$, in network $m$ in every time window of 20ms by using a population vector. We used this collection of samples to estimate the mean, $\langle \tilde{\mathbf{s}} \rangle = (\langle \tilde{s}_1 \rangle, \langle \tilde{s}_2 \rangle)^\top$, and covariance matrix, $\mathbf{\Sigma_s}$, of the sampling distribution. Meanwhile, the mean $\boldsymbol{\mu}_f$ and precision matrix $\boldsymbol{\Lambda}_f$ of the likelihood are linearly read out from the feedforward inputs fed into the network model (Eq. (33)).

If the sampling distribution is comparable with the posterior, the sampling mean $\langle \tilde{\mathbf{s}} \rangle$ and covariance $\mathbf{\Sigma_s}$ should satisfy Eq. (34). We use the actual sampling covariance and the likelihood parameters to predict the sampling mean, i.e., $\langle \tilde{\mathbf{s}} \rangle_{\text{pred}} = \mathbf{\Sigma_s}\boldsymbol{\Lambda}_f\boldsymbol{\mu}_f$, and compare it with the actual $\langle \tilde{\mathbf{s}} \rangle$ (Fig. 7d–f). To obtain the posterior precision matrix, given the sampling mean $\langle \tilde{\mathbf{s}} \rangle$ and the likelihood parameters, we vary the single parameter of prior precision $\Lambda_s$ to minimize the KL divergence from the prediction of posterior by using the value of $\Lambda_s$, and the actual sampling distribution. Given this value of $\Lambda_s$, the prediction of posterior precision is computed as $\mathbf{K}_{\text{pred}} = \boldsymbol{\Lambda}_s + \boldsymbol{\Lambda}_f$ (Eq. (34)) which is then compared with actual sampling precision matrix ($\mathbf{\Sigma_s}^{-1}$; see Fig. 7c–g). The prior precision, $\Lambda_s$, is a *subjective* prior, which reflects the prior stored in the recurrent network and may change with input (see Discussion). More details of network simulation and parameters can be found in Supplementary Information.

### Reporting summary
Further information on research design is available in the Nature Portfolio Reporting Summary linked to this article.

## Data availability
This is a strictly computational study and all data used in making figures were generated by computer simulations of the proposed model with the link of codes shown in Code Availability.

## Code availability
The code of network simulation was written in MATLAB 2018b, and can be found at GitHub (https://github.com/wenhao-z/Sampling_PoissSpk_Neuron)[73].

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

## Acknowledgements

National Institutes of Health grants 1R01MH115557 (K.J.), 1U19NS107613-01 (B.D.), R01EB026953 (B.D.), R01EY034723 (B.D.); National Science Foundation grant NSF-DBI-1707400 (K.J.), DMS-2207647 (K.J.). Vannevar Bush faculty fellowship N00014-18-1-2002 (B.D); Simons Foundation Collaboration on the Global Brain (B.D.).

## Author contributions

W.H.Z., S.W., K.J., and B.D. conceived and designed the study. W.H.Z. developed the ideas, performed the analyses, and the numerical simulations. W.H.Z, K.J., and B.D. discussed the results and wrote the manuscript.

## Competing interests

The authors declare no competing interests.
