## [Peer Review File · Nature Communications]

Sampling-based Bayesian inference in recurrent circuits of stochastic spiking neuronsREVIEWER COMMENTS

Reviewer #1 (Remarks to the Author):

The manuscript presents a theory of how recurrent circuitry and spiking variability could contribute to an important computational goal of cortical circuits, namely probabilistic inference. The manuscript combines two key insights. First, if properly tuned, recurrent connections can store an internal model of sensory signals, a Bayesian prior. Second, spiking variability provides the means to sample from the inferred posterior distribution (which combines the likelihood/sensory evidence conveyed by feedforward inputs, with the prior conveyed by recurrent inputs), therefore implementing a sampling-based representation of uncertainty. The authors design networks where both the likelihood and prior are represented with probabilistic population codes (PPC) and therefore simple linear combinations of neural activity result in the correct (log-)posterior over the stimulus variable, which is then sampled by the network's variable activity. This is different from most literature on neural sampling focused on inference over low-level latent variables (eg. wavelet coefficients of images). The authors show that the recurrence needs to be tuned precisely so that the network samples the correct posterior, and that this implies that noise correlations (here induced only by the recurrence, because the inputs are assumed independent Poisson) are determined by the prior stored in the circuit. The authors illustrate these ideas with simulations using two different generative models of a stimulus plus a context, one hierarchical and one parallel. Lastly, the manuscript shows that recurrence induces differential noise correlations, and offers the prediction for how the amplitude of differential correlations and task performance should change with perceptual learning. Overall this is an interesting manuscript that contributes to understanding the circuit mechanisms of probabilistic inference. I also find there are some moderate weaknesses.

On the technical side, the derivations and simulations appear all correct. Two conceptual issues could be addressed better in my opinion. First, the choice of the generative model of figure 3: the "context" variable in that case is not really context, the stimulus $s = z + \text{noise}$ so the inference is trivial. This is very different from the kind of model that Fig 3 alludes to, and so the figure and the word "context" here are misleading. The second generative model is more interesting. But in both cases I would suggest clarifying earlier on what is the advantage of using Poisson noise to generate samples, given that a noiseless network would provide the instantaneous posterior of s_1 (because of the PPC design): if I understood correctly, the main advantage of using a sampling based representation is that it draws samples from the joint 2D posterior and so marginalization is trivial. These are general advantages of sampling-based codes, but may be worth mentioning more clearly.

The second and main conceptual issue, that I think requires more discussion, is that the optimal recurrent weights (i.e. required to sample from the correct posterior) depend on the input. This means that the network could not sample the correct posterior when the stimulus changes to higher or lower uncertainty. Only around line 390 this issue is addressed, and only briefly in two sentences and figure 7c. This shows that even with fixed recurrent weights, changing input uncertainty leads to a change in sampling precision qualitatively consistent with the input. But as the equations show clearly, the sampled posterior cannot be the correct one. Because a calibrated representation of uncertainty is

central to probabilistic inference, this issue deserves more space and better quantification. One possible solution for example would be to consider optimizing the recurrence for an “ecological” family of distribution (rather than the optimal posterior for a specific input), which would require a more general objective function, and then quantify the information loss across uncertainty levels. With this choice, one would have to think also about the implications for the predictions on differential correlations.

The manuscript does a good job, in particular in Discussion, placing this work in context and distinguishing it from abundant related literature, with a couple of exceptions. In particular, the key aspects of the proposed models, i.e. 1) tuning the recurrent weights and 2) the role of spiking variability and defining a sample as a linear combination of population activity, and 3) learning and differential correlations, have been central also to, respectively, 1) Echeveste et al 2020 and 2) Savin & Deneve 2014 (and work from the Maas lab), and 3) Lange & Haefner. This reduces a bit the innovation of this manuscript, which appears more of another variation on the theme of how sampling-based models capture aspects of neural activity (here differential correlations).

A more minor issue is that the main success of the sampling-based models cited has been that they capture many aspects of cortical physiology, including modulations of response variability, covariability, dynamics and oscillations when uncertainty changes. In this manuscript, with the exception of differential correlations, it is not clear if the networks generate activity consistent with those empirical data.

MISC

Line 30 “variability in spike generation” confuses me. Isn’t the spike generation almost noiseless (Mainen & Sejnowski 1995), and variability due instead mainly to chaotic network dynamics?

Line 38 ref [12] should be [11]?

Line 88 “inhibitory activity acts to stabilize” is this true also with linear transfer function?

Line 122 Henaff et al 2020 Nature Communications and Festa et al 2021 Nature Communications seem relevant too here for empirical support

Line 450 typo “Fig 8a”

Line 653 typo “imples”

Line 683 typo “ththe”

Reviewer #2 (Remarks to the Author):

The authors introduce a neuronal network with realistic noise and connectivity that can sample from an underlying probability distribution and perform Bayesian inference.

The paper builds on a large body of research on probability representations of stimuli and sampling probability distribution from them using neural networks. The paper adds Poisson variability into the picture and provides novel theoretical results, but the results are not very surprising in view of the previous literature. Several works have shown already how to sample probability distributions using variability and neural networks, and some of them are quoted by the authors.

Moreover, some of the approximations made theoretically do not seem to be well justified. For instance, the authors approximate the instantaneous rate through a Gaussian profile, but this instantaneous rate enters in an equation where its evolution will be noisy. More specifically, in eq. 10 the instantaneous rate depends on the previous history of spikes, so it is unclear how the instantaneous rate can obey a Gaussian profile at all time instances, as it is assumed in eq. 12. Maybe I am missing something. Further, eq. 10 does not seem to be linear, because instantaneous rates cannot be negative and Gaussian noise can be large and negative. If noise is small, it can be a good approximation, but not in general. Further, if the noise is white (eq. 11), then the rate will be negative half of the time bins.

Finally, although the recurrent connections store an internal model of the external world, the connections need to be tuned to the stimuli and context and their correlations, which seems quite unrealistic in scenarios with varying stimuli and contexts.

-“remains a mystery” in line 25 maybe is too much due to the extensive literature on the topic of variability

-lines 109-10. The rhetoric question seems to be far-fetched, as it is likely that a role of recurrent connections is to perform non-linear computations for complex input-to-output transformations.

We thank the reviewers for their comprehensive critique of our study. We have attempted to address their concerns, and we now feel that our resubmitted manuscript has improved considerably. We address each reviewer's concerns below. Reviewer comments are in black while our responses are in blue.

Reviewer #1:

The manuscript presents a theory of how recurrent circuitry and spiking variability could contribute to an important computational goal of cortical circuits, namely probabilistic inference. The manuscript combines two key insights. First, if properly tuned, recurrent connections can store an internal model of sensory signals, a Bayesian prior. Second, spiking variability provides the means to sample from the inferred posterior distribution (which combines the likelihood/sensory evidence conveyed by feedforward inputs, with the prior conveyed by recurrent inputs), therefore implementing a sampling-based representation of uncertainty.

The authors design networks where both the likelihood and prior are represented with probabilistic population codes (PPC) and therefore simple linear combinations of neural activity result in the correct (log-)posterior over the stimulus variable, which is then sampled by the network's variable activity. This is different from most literature on neural sampling focused on inference over low-level latent variables (e.g. wavelet coefficients of images).

We thank the reviewer for their succinct summary of our work. Indeed, one of our goals was to describe a biologically feasible implementation of sampling that is extendable to more complex, hierarchical generative models.

The authors show that the recurrence needs to be tuned precisely so that the network samples the correct posterior, and that this implies that noise correlations (here induced only by the recurrence, because the inputs are assumed independent Poisson) are determined by the prior stored in the circuit. The authors illustrate these ideas with simulations using two different generative models of a stimulus plus a context, one hierarchical and one parallel. Lastly, the manuscript shows that recurrence induces differential noise correlations, and offers the prediction for how the amplitude of differential correlations and task performance should change with perceptual learning. Overall this is an interesting manuscript that contributes to understanding the circuit mechanisms of probabilistic inference.

We agree with this summary, and believe the prediction surrounding differential correlations is another novel contribution of our work. The tuned recurrent weights provide a way to *analytically* relate connectivity, population response variability, and the generative model used for Bayesian inference. We also make experimentally testable predictions about the relation between these three quantities.

Our work links these three properties of neural circuits, *i.e.*, their structure, the statistics of their response, and the inference algorithm they perform (see Savin NeurIPS 2014 for another, distinct approach). The relations among the above three properties, by their very nature, make the model parameters appear to be 'fine-tuned'. However, although the recurrent weights need to be tuned, we only tune the overall recurrent weight strength (a scalar) and not the individual neuron-to-

neuron weights in the coupling matrix. This significantly reduces the complexity of the analysis, making it possible to derive interpretable relationships between the coupling strengths, the prior represented by this coupling, and the resulting covariability in the response. Similar parameterizations and simplifications were also used in previous studies, e.g., Eq. 10 in Echeveste et al., Nat. Neurosci., 2020. Indeed, we believe that our analysis provides some further insights into the results of previous, largely computational studies.

In addition, to clarify our network doesn't need precise tuning of recurrent weights we have added and revised the following text in the Discussion (lines 574-581):

“Moreover, the recurrent connections in our network model are translation-invariant in the stimulus subspace, an assumption widely used in studies of continuous attractor networks (CAN) [22, 56, 59, 60], and a recent network model implementing sampling [32]. Perfectly translation-invariant connections are not strictly required in a circuit implementing sampling, but this assumption allows us to simplify the mathematical analysis. Adding randomness in recurrent connectivity would increase the variance of sampling distributions. We could then adjust the overall recurrent weight (a scalar) so that the sampling distribution matches the posterior, with no need to fine-tune individual synaptic weights in the network model.”

I also find there are some moderate weaknesses. On the technical side, the derivations and simulations appear all correct. Two conceptual issues could be addressed better in my opinion. First, the choice of the generative model of figure 3: the “context” variable in that case is not really context, the stimulus $s=z+\text{noise}$ so the inference is trivial. This is very different from the kind of model that Fig 3 alludes to, and so the figure and the word “context” here are misleading. The second generative model is more interesting.

We agree that the hierarchical generative model shown in Fig. 3 is simple because the stimulus s is just a noisy version of the “context” z . Moreover the context, z , typically influences the distribution of multiple stimuli, s , at different spatial locations. In contrast, we only consider one local stimulus rather than many. We do so to simplify the initial analysis and presentation. We start with this simplified generative model to introduce the essential ideas behind the computation, and the mechanisms that also drive sampling in more complex situations. To clarify this point, we now refer to the z variable as the “stimulus parameter”, and have revised the manuscript accordingly. We emphasize that the stimulus parameter z is a deeper latent variable than the stimulus s , and this point is essential for the extensions to more complex generative models.

The reviewer is correct that the generative model in Fig. 3a is somewhat trivial, however we believe that it plays an important didactic role in our manuscript. Our study does not just focus on the neural circuit implementation of sampling given this simple hierarchical generative model, but builds on it to describe neural implementation for inference under three different generative models of increasing complexity (Fig. 2a, Fig. 3a, and Fig. 6a). The first model (Fig. 2a) describes the mechanisms of inference and sampling in the simplest possible setting. The generative models in Fig. 2a and Fig. 3a differ in the inclusion of stimulus parameter, z . Without a stimulus parameter z , the optimal sampling network is a feedforward network (Fig. 2a). Adding the stimulus parameter, z , complicates the second model, and requires the addition of

recurrent connections in the network model (i.e., recurrent inputs represent a sample of z), and a more careful analysis. Finally, the third model involves a pair of stimuli s_1 and s_2 , and a pair of recurrently coupled networks are required. These three generative models form the basic building blocks of large-scale generative models that are used in practical applications. As we now clarify in the text, finding the neural circuit implementation for inference under these basic generative models can help us build biologically plausible neural network models and mechanisms for inference in more complex situations.

We have revised the manuscript accordingly by

- 1) Replacing the wording “context” by “stimulus parameter” throughout the manuscript.
- 2) Adding new text in the first paragraph in Discussion (lines 481-486) to emphasize the significance of understanding neural sampling under these three generative models:

“We have shown how sampling can be implemented using biologically feasible mechanisms for three different generative models of increasing complexity. The simplest model includes one latent stimulus, while the more complex models include multiple latent stimuli organized hierarchically or in parallel. These three generative models form the basic building blocks of more complex models. Thus our ideas can be extended to a wide range of perceptual and cognitive processes [49].”

We believe that of particular importance are examples of generative models represented by the network not matching the world structure. Such generative models may not capture the full complexity of latent stimuli and their relations with measured stimuli. It has been shown that in such cases network responses can be biased, or more noisy than expected if using the true generative model. Understanding such suboptimal inference in networks that implement inference via sampling would allow us to better relate our findings to psychophysics. Although this is beyond the scope of the present manuscript, we now mention this point in the Discussion (lines 603-607):

“Here we assumed that the generative model represented by the network matches the model that generate the sensory stimuli. This is unlikely to be the case in practice. Such mismatch between the true and internal model of the world can lead to biases and increased noise which are likely to manifest in specific ways in neural circuits that perform inference via sampling [67].”

But in both cases I would suggest clarifying earlier on what is the advantage of using Poisson noise to generate samples, given that a noiseless network would provide the instantaneous posterior of s_1 (because of the PPC design): if I understood correctly, the main advantage of using a sampling based representation is that it draws samples from the joint 2D posterior and so marginalization is trivial. These are general advantages of sampling-base codes, but may be worth mentioning more clearly.

We agree with this suggestion. In the revised manuscript we emphasize the motivation for, and the computational advantages of using Poisson sampling in the main text and provide further details in the Discussion. We now motivate and discuss the advantages of Poisson sampling in more detail at several points in the revised text:

Motivation of Poisson spiking sampling

An important reason for considering Poisson spiking variability for sampling is that such variability is often observed in biological networks. Moreover, Poisson-like variability can be internally generated using relatively well understood mechanisms such as chaotic spiking network dynamics, as suggested by the reviewer. We explain the motivation for using Poisson variability to drive sampling in the following new paragraph (lines 126-132):

“Implementing sampling requires a network that generates variable output with stable statistics. It has been well documented that cortical spiking responses are often approximately Poissonian [3, 35]. Theoretical studies suggest that such Poissonian variability can be internally generated in a network with dynamically balanced recurrent excitation and inhibition [36, 37]. We thus assumed that our model neurons are Poissonian, and used the resulting fluctuations as the internal source of variability needed for sampling-based Bayesian inference.”

We also added the following possible caveat right after above text to the Discussion (lines 132-136):

“It remains to be shown if discrete Poissonian variability can be used to generate samples from stimuli with continuous probability distributions (e.g., orientation, moving direction) with the flexibility needed to represent different stimulus uncertainties. However, spike counts are discrete. It is thus possible that the errors that result from such representations are characteristic of stimulus inference by animals that use sampling.”

The advantage of Poisson spike sampling in multivariate case

We agree that the sampling is trivial in the case of the *univariate* posterior shown in Fig. 2a. In this case samples are drawn directly from the posterior (the conditional distribution in Eq. 1 is the posterior), and a deterministic network can represent the univariate posterior by a single time response avoiding the costly process of collecting samples (as discussed in lines 187-191 in the original manuscript). As we noted in our answer to a previous comment, we introduced the example in Fig. 2a to introduce the main ideas behind our theory, and to make the rest of the manuscript easier to follow.

In the case with *multivariate* posteriors (Fig. 3a and Fig. 6a), even if population *inputs* are modeled using probabilistic population codes (PPCs), sampling greatly simplifies the neural circuit implementation (in addition to allowing for easy marginalization). For example, to infer the 2D posterior under the generative model in Fig. 6a, a deterministic network faces two main challenges that increase the complexity of the neural implementation. First, a 2D posterior can be linearly represented in a *linear* deterministic network whose output *spikes* are also described by the PPC, but the number of neurons in the network will be N^2 compared to the $2N$ neurons in our implementation. Second, the number of neurons in linear and deterministic networks will increase *exponentially* with the latent stimulus dimension (as shown by Fiser et al., Trends Cog. Sci., 2010). The size of our network increases linearly with the number of feature dimensions. Another possibility is that the 2D posterior is also represented by two coupled deterministic

networks with a total of $2N$ neurons, with structure similar to the one we describe in our work (Fig. 6a). However, in this case the deterministic network model needs to perform complicated nonlinear operations (Raju, NeurIPS 2016, Fig. 3; as discussed at line 511 in our original manuscript). In contrast, the implementation we propose results in a linearly coupled network model that can sample from and represent of 2D posterior distributions.

Another reason that Poisson sampling is non-trivial in the multivariate case is that Eqs. (4b-4c; 8b) describe how the network generates samples from *conditional distributions*, rather than directly from the posteriors. That is, in the cases described by Eqs. (4b-4c; 8b) no instantaneous variables in network dynamics directly represent the posterior, and the posterior is only represented by samples over time.

In accord with these points, we revised the Discussions in the manuscript (lines 503-511) and quote the relevant text below:

“Although it takes time to collect sufficiently many samples to approximate the posterior well, an advantage of sampling codes compared to PPCs is that inference with multivariate posteriors can be implemented using linearly coupled subnetworks (Fig. 6), with the number of subnetworks determined by the dimension of the latent stimulus features. In contrast, to represent an M -dimensional multivariate posterior using PPCs requires N^M neurons in a linear network (N is the number of neurons in representing each dimension) so that the number of neurons increases exponentially with the latent stimulus dimension, M [16]. Alternatively, coupled networks with NM neurons can be used, but require complex, nonlinear coupling between these networks [50, 51].”

The second and main conceptual issue, that I think requires more discussion, is that the optimal recurrent weights (i.e. required to sample from the correct posterior) depend on the input. This means that the network could not sample the correct posterior when the stimulus changes to higher or lower uncertainty. Only around line 390 this issues is addressed, and only briefly in two sentences and figure 7c. This shows that even with fixed recurrent weights, changing input uncertainty leads to a change in sampling precision qualitatively consistent with the input. But as the equations show clearly, the sampled posterior cannot be the correct one. Because a calibrated representation of uncertainty is central to probabilistic inference, this issue deserves more space and better quantification. One possible solution for example would be to consider optimizing the recurrence for an “ecological” family of distribution (rather than the optimal posterior for a specific input), which would require a more general objective function, and then quantify the information loss across uncertainty levels. With this choice, one would have to think also about the implications for the predictions on differential correlations.

We thank the reviewer for bringing up this an important issue, and we now point out the dependency of input on the optimal recurrent weight in our network earlier in the manuscript (lines 273-277):

“Moreover, the optimal recurrent weight also depends on the likelihood precision, Λ_f , that is determined by the input spike count. Hence, the optimal weight needs to be adjusted depending

on feedforward inputs so that the network generates samples from the correct posterior (see Discussion of how this feature impacts network sampling)."

In addition, we have added the following to the Discussion on lines 593-603 to describe a way to optimize the recurrent weights for an ecological family of distributions, as suggested by the reviewer:

"One possibility is that the proposed network model does not generate samples from each distinct posterior determined by a specific feedforward input, $p(s|u^f)$, but rather generates samples from the average sampling distribution over all possible feedforward inputs and hence matches the average posterior distribution $E_{p(u^f)}[p(s|u^f)] = E_{p(u^f)}[q(s|u^f)]$, where $E_{p(u^f)}[\cdot]$ denotes the average over the distribution $p(u^f)$. Since the proposed recurrent circuit is general, this result may explain one source of inductive bias in cortical processing [62]. On the other hand, sampling correctly from each specific posterior could be achieved using different biophysical mechanisms that can modulate synaptic strengths and that we have not included in our model. For instance, short-term synaptic depression [63] or spike frequency adaptation [64] are gain control mechanisms that would allow the recurrent input strength (representing the prior correlation) to remain relatively fixed despite an increase in the feedforward input strength."

The manuscript does a good job, in particular in Discussion, placing this work in context and distinguishing it from abundant related literature, with a couple of exceptions. In particular, the key aspects of the proposed models, i.e. 1) tuning the recurrent weights and 2) the role of spiking variability and defining a sample as a linear combination of population activity, and 3) learning and differential correlations, have been central also to, respectively, 1) Echeveste et al 2020 and 2) Savin & Deneve 2014 (and work from the Maas lab), and 3) Lange & Haefner. This reduces a bit the innovation of this manuscript, which appears more of another variation on the theme of how sampling-based models capture aspects of neural activity (here differential correlations).

We thank the reviewer for pointing to these previous studies which introduce related ideas. One way our work differs from those studies is that we show how (discrete) spiking neural responses can be used to generate samples from the posterior of a continuous variable. Most previous studies implicitly approximate discrete spiking variability by a Gaussian distribution. However, this approximation may not work well when only few spikes are emitted. When firing rates are low, or the distribution of spike counts is not Gaussian. In contrast, we propose that samples from continuous distributions using Poissonian neurons can be achieved using the (temporally averaged) smooth population firing rate profile. Our approach can work well even if only a few spikes are emitted across the whole neuronal population. We now emphasize this point in the Discussion (lines 512-519):

"Neurons emit a discrete number of spikes, but their responses often need to represent continuous quantities. Most studies of neural sampling implicitly rely on approximating Poissonian spike counts with Gaussian variables (e.g., [29, 31, 50]). However, this approximation does not work well when only a few spikes are emitted. Here, we showed that discrete Poisson spike generation can be used to generate samples from a posterior distribution of a continuous stimulus feature using a temporally averaged, smooth population firing rate

profile. Thus, we have shown how a sample from a continuous variable can be generated even with only a few spikes from the neuronal population.”

A more minor issue is that the main success of the sampling-based models cited has been that they capture many aspects of cortical physiology, including modulations of response variability, covariability, dynamics and oscillations when uncertainty changes. In this manuscript, with the exception of differential correlations, it is not clear if the networks generate activity consistent with those empirical data.

In Fig. S1 in the original manuscript, we showed that the network can produce responses that are qualitatively similar to experimental observations. In particular, this includes the dependence of spike count variance (Fig. S1B) and the Fano factor of the neural response (Fig. S1C) on the difference between preferred and presented stimulus orientation. The model also captures the decay of pairwise noise correlations with the difference in preferred orientations observed experimentally (Fig. S1D). In the revised manuscript we add a sentence in the main text to better describe these comparisons (line 92-94).

MISC

Line 30 “variability in spike generation” confuses me. Isn’t the spike generation almost noiseless (Mainen & Sejnowski 1995), and variability due instead mainly to chaotic network dynamics?

Our network is agnostic with respect to the actual mechanisms underlying the Poisson spike generation in our network. However, it is reasonable to assume that the variability used in our study is a result of chaotic spiking network dynamics, i.e., it is a network effect rather than due to internal noise in single neurons. As the reviewer points out, channel noise, and other sources of internal fluctuations likely have a small impact on the variability of neural responses. In an ongoing follow-up to this study we will show that noisy responses in chaotic, balanced networks can indeed provide the desired population variability, and that such networks can implement the type of inference we describe. The wording “variability in spike generation” follows a number of previous studies that used Poisson neurons to fit neural data, such as the work of J. Pillow, L. Paninski and E. Simoncelli, and their collaborators.

Line 38 ref [12] should be [11]?

We have fixed this typo. Thanks!

Line 88 “inhibitory activity acts to stabilize” is this true also with linear transfer function?

Yes. Without inhibitory neurons a network of the type we describe with recurrent weights between excitatory neurons will be unstable, even if the transfer function is linear.

Line 122 Henaff et al 2020 Nature Communications and Festa et al 2021 Nature Communications seem relevant too here for empirical support

Thank you for pointing out these references. We include them in the revised manuscript (lines 124 and 520).

Line 450 typo “Fig 8a”

Line 653 typo “imples”

Line 683 typo “ththe”

We have fixed these typos in the revised manuscript.

Reviewer #2:

The authors introduce a neuronal network with realistic noise and connectivity that can sample from an underlying probability distribution and perform Bayesian inference.

The paper builds on a large body of research on probability representations of stimuli and sampling probability distribution from them using neural networks. The paper adds Poisson variability into the picture and provides novel theoretical results, but the results are not very surprising in view of the previous literature. Several works have shown already how to sample probability distributions using variability and neural networks, and some of them are quoted by the authors.

We agree that several previous studies have addressed similar questions. Yet, we believe that our results differ from those in previous work in important ways. We have revised the manuscript to better explain these findings, and show how they differ from and complement previous work (see response to reviewer 1, and the text we quote below).

Our manuscript differs from previous studies in several ways.

- 1) We derive a biologically plausible neural circuit implementation for sampling from multivariate posteriors (e.g., Fig. 6a). How to generate samples and perform inference using a physiologically plausible implementation has not been well understood when posteriors are multivariate because most of previous network sampling studies focus on sampling the posterior of a univariate stimulus feature. Moreover, the representation of multivariate posterior in the framework of probabilistic population codes requires more complicated circuit than ours (see Discussions at lines 503-511). Moreover, the proposed implementation provides experimentally testable hypotheses about the origin and behavior or “differential correlations”.
- 2) We derive relationships between the generative model, neural circuit dynamics, and neuronal response variability, providing further concrete, experimentally testable predictions. Specifically, we give a quantitative relation among the magnitude of internally generated differential correlation, prior correlation, and the recurrent connectivity which is summarized in Eq. 9.
- 3) As noted by reviewer #1, the generative models in our study differ from those considered previously in the literature on neural sampling which is focused on inference of low-level latent variables, e.g. decomposing an image into its wavelet coefficients (e.g., Echeveste et al., 2020; Orban et al., 2016; Haefner et al., 2016; Aitchison et al., 2016; Savin et al., 2014; Buesing et al., 2011; Hoyer, 2003; Festa et al 2021). In contrast, we consider a latent stimulus feature(s) such as

orientation, heading direction, which allows us to use a continuous attractor network for inference. While similar generative models were considered previously, especially in the behavioral studies of (multisensory) cue integration, (e.g., Ma et al., 2006; Hillis et al., 2002; Shams et al., 2005; Bresciani et al., 2006; Roach et al., 2006), those studies did not consider neural sampling and/or did not focus on the neural circuit implementation of the proposed inference algorithms. Thus, our work fills a gap in our understanding of how a recurrent network of neurons can generate samples from distributions of an often-used description of latent stimuli which is supported by behavioral data. Our study investigates the network sampling of latent variables organized in parallel or hierarchically. These generative models form the basic building blocks of more complex models. Thus, the mechanism we describe is generalizable to more complicated generative models.

4) We describe a novel mechanism for using discrete Poisson spiking variability to generate samples from different types of continuous distributions (determined by the population firing rate profile). This is in contrast with previous studies which approximate Poissonian counts as Gaussian variables. We added the following text to the revised manuscript to emphasize this point (see lines 512-519 in the revised manuscript):

“Neurons emit a discrete number of spikes, but their responses often need to represent continuous quantities. Most studies of neural sampling implicitly rely on approximating Poissonian spike counts with Gaussian variables (e.g., [29, 31, 50]). However, this approximation does not work well when only a few spikes are emitted. Here, we showed that discrete Poisson spike generation can be used to generate samples from a posterior distribution of a continuous stimulus feature using a temporally averaged, smooth population firing rate profile. Thus, we have shown how a sample from a continuous variable can be generated even with only a few spikes from the neuronal population.”

Poisson spiking variability can be internally generated in chaotic spiking networks, as also noted by reviewer 1. In future work we will illustrate how chaotic spiking networks can implement sampling via internally generated variability, i.e., without invoking an abstract random number generator. We think this problem has not been addressed in past sampling studies. We now emphasize this point in the revised manuscript by adding a further explanation (lines 126-132):

“Implementing sampling requires a network that generates variable output with stable statistics. It has been well documented that cortical spiking responses are often approximately Poissonian [3, 35]. Theoretical studies suggest that such Poissonian variability can be internally generated in a network with dynamically balanced recurrent excitation and inhibition [36, 37]. We thus assumed that our model neurons are Poissonian, and used the resulting fluctuations as the internal source of variability needed for sampling-based Bayesian inference.”

Moreover, some of the approximations made theoretically do not seem to be well justified. For instance, the authors approximate the instantaneous rate through a Gaussian profile, but this instantaneous rate enters in an equation where its evolution will be noisy. More specifically, in eq. 10 the instantaneous rate depends on the previous history of spikes, so it is unclear how the

instantaneous rate can obey a Gaussian profile at all time instances, as it is assumed in eq. 12. Maybe I am missing something.

This is an important point: Indeed, in the initial derivation we assume that the instantaneous population firing rate is smooth (Eq. 12) to simplify our analysis, and help readers understand the main ideas behind the derivation. The actual instantaneous firing rate will not be smooth, as the reviewer correctly points out. We show in Eqs. 17 and 39 that a noisy instantaneous population firing rate will not affect our derivation substantially, since most of fluctuations are orthogonal to the directions that determines the values of the samples. In numerical simulations the instantaneous firing is not smooth, and the simulation confirm that the whole neural circuit can still generate samples from the posterior with sufficient accuracy.

To clarify this point, we have added the following text early in the Results (lines 156-159) section to point out that our theory works even when the instantaneous firing rates are noisy:

“We assumed that instantaneous population firing rates are smooth to simplify the analysis, but this assumption is not essential. Poissonian variability can be used to generate samples from the proper distribution as long as the temporally averaged population firing rate is smooth, even if the instantaneous population firing rate is noisy (see Eq. 17).”

Further, eq. 10 does not seem to be linear, because instantaneous rates cannot be negative and Gaussian noise can be large and negative. If noise is small, it can be a good approximation, but not in general. Further, if the noise is white (eq. 11), then the rate will be negative half of the time bins.

We apologize for the confusion in our initial description of our model. Eq. 10 is linear since we did not add an explicit negative rectification of the firing rate which means the instantaneous firing rate, λ_t , can be negative in numerical simulations due to negative recurrent input u_t^r and the noise. The “rectification” comes at the Poisson spike generation (Eq. 11) where we assumed that a negative λ_t implies a zero probability in generating a spike. Thus, as the reviewer notes, our model in Eq. (10) and (11) is nonlinear. We clarify this point by adding the following text after Eq. 10 in Methods (lines 651-652):

“The instantaneous firing rate λ_t can be negative due to the recurrent input and noise (Eq. 36). We interpret a negative firing rate, λ_t , as a zero probability of generating a spike.”

Moreover, we only use recurrent noise (a non-zero σ_r in Eq. 10) in the network model that generates samples from the posterior distribution described in Fig. 3a. In this case, we used multiplicative recurrent noise to approximate Poisson variability in order to generate samples of the context, z . In contrast, recurrent noise is not included in the network model that generates samples from the posterior distribution corresponding to the models shown in Fig. 2a and Fig. 6a (i.e., $\sigma_r = 0$, in Eq. 36). Overall, the inclusion of recurrent noise does not affect the main mechanism we describe which is based on using Poisson spiking variability to generate samples from continuous stimulus distributions.

Finally, although the recurrent connections store an internal model of the external world, the connections need to be tuned to the stimuli and context and their correlations, which seems quite unrealistic in scenarios with varying stimuli and contexts.

We agree that in some situations the stimuli and context will vary since the world is in constant flux. In this case the dynamic latent variables can be described by a hidden Markov model (HMM), and our network model has the potential to infer the hidden states in such an HMM by generating samples so that the instantaneous firing rate represents the conditional distribution of the latent stimulus given all previous feedforward inputs, i.e., $p(s_t | \{u_n^f\}_{n=0}^t)$, and the emitted Poisson spikes represent a sample from this distribution. We added a sentence in Discussion to introduce this possibility (lines 627-628):

“And the proposed network model has the potential to produce samples from the posterior distribution of latent dynamic stimuli which can be described by a hidden Markov model.”

The question of inference when the correlations between latent stimuli or latent stimulus and context are dynamic is harder. We can speculate that the temporal dynamics can modulate the effective connectivity within the network, or that the connectivity can be modulated by top-down inputs. We only need to tune the overall recurrent weight (a scalar) and the recurrent weight matrix satisfying the translation-invariant structure on the stimulus feature subspace (see lines 574-581 in the revised manuscript). We believe that the overall structure of the network that reflects the statistical regularity of the natural world (e.g., translation invariance) could have been learned over evolutionary timescale, and may be fine-tuned during development. The tuned recurrent weights provide a way to relate connectivity, population response variability, and Bayesian inference.

-“remains a mystery” in line 25 maybe is too much due to the extensive literature on the topic of variability

We agree, and have revised the text accordingly (see line 25 in the revised manuscript).

-lines 109-10. The rhetoric question seems to be far-fetched, as it is likely that a role of recurrent connections is to perform non-linear computations for complex input-to-output transformations.

We agree, and have revised the text accordingly (see lines 111-112).

REVIEWERS' COMMENTS

Reviewer #1 (Remarks to the Author):

The authors have addressed my concerns, and I find the manuscript improved. In particular, the text discusses more clearly the issue of input-dependent recurrent weights needed for exact posteriors.

As far as I can tell, the authors also provide valid answers to the comments of the other reviewer.

Two details:

- Fig 3A, the word 'context' should be replaced with 'stimulus parameter' consistent with the revised text

- Many links to equations and figures are missing in the main and supplementary pdf (search for '??')

Reviewer #2 (Remarks to the Author):

I appreciate the improvements that the reviewers have added to the paper. However, I think that building a realistic neural network that can only sample a multivariate Gaussian random variable, or similar, does not seem to be a very important step forward. More important questions not addressed are how to realistically sample more complex posteriors, and how they can be learned in changing environments.

REVIEWERS' COMMENTS

Reviewer #1 (Remarks to the Author):

The authors have addressed my concerns, and I find the manuscript improved. In particular, the text discusses more clearly the issue of input-dependent recurrent weights needed for exact posteriors.

As far as I can tell, the authors also provide valid answers to the comments of the other reviewer.

Thanks very much for the reviewer positive comments on our revised manuscript.

Two details:

- Fig 3A, the word 'context' should be replaced with 'stimulus parameter' consistent with the revised text

This has been revised.

- Many links to equations and figures are missing in the main and supplementary pdf (search for '??')

It is due to an error in compiling the LaTeX file. We will proofread the revised manuscript.

Reviewer #2 (Remarks to the Author):

I appreciate the improvements that the reviewers have added to the paper. However, I think that building a realistic neural network that can only sample a multivariate Gaussian random variable, or similar, does not seem to be a very important step forward. More important questions not addressed are how to realistically sample more complex posteriors, and how they can be learned in changing environments.

We thank the reviewer for their constructive comments on our revised manuscript. We cannot agree more that sampling complex posterior and learning in changing environments are important questions which will also form our future research, e.g., we are studying the neural circuit sampling of mixture distributions. However, the mechanistic insights provided in this study are critical to build networks that can sample more complex probabilistic stimuli.